



# Drop clustering and drop size correlations from holographic imagery suggest cloud droplet spectral broadening via entrainment-mixing

John J. D'Alessandro[1,*], Robert Wood[1], Peter N. Blossey[1]

[1]Department of Atmospheric Sciences, University of Washington, Seattle, WA, USA
[*]Current Affiliation: Laboratoire d'Optique Atmosphérique, Université de Lille, Lille, Halle du Nord, France

*Correspondence to*: John J. D'Alessandro (john.d-alessandro@univ-lille.fr)

**Abstract.** The question of how droplets rapidly grow large enough to initiate collision-coalescence has persisted for decades. Many theories suggest drop clustering on millimeter scales can produce sufficiently large drops (i.e., those in the "bottleneck" size range; ~25 to 50 µm diameters).

A novel method is introduced to evaluate drop clustering trends particle-by-particle (i.e., the number/proximity of neighboring drops for given droplets, defined as drop clustering fields)—in contrast to previous studies which determine drop clustering metrics of given sample volumes. Specifically, this study evaluates the statistical likelihood that drops of a given size will either be associated with a significant number of neighboring drops, or be significantly isolated from neighboring drops.

Observations are acquired from the HOLODEC during the Cloud System Evolution in the Trades campaign, which sampled subtropical marine warm clouds. The HOLODEC measures drop size distributions and the 3D spatial coordinates of droplets. Results show drops within the bottleneck size range (diameters of ~25–50 µm) are most likely to be significantly isolated from neighboring drops. This "isolated large drop trend" is primarily observed at subsaturated conditions, suggesting entrainment is the contributing factor. Holograms associated with this trend are more likely to have greater mean diameters, suggesting smaller drops are preferentially evaporating. However, these holograms are also more likely to have broader drop size distributions, larger maximum drop sizes and overly regions where precipitation reaches the lowest altitudes from the sampled cloud, suggesting entrainment-mixing drop size distribution broadening is a relevant precipitation-initiation mechanism.

## 1 Introduction

Explanations for the rapid production of precipitation in warm clouds (e.g., Rauber et al. 2007; Saunders 1965) have eluded researchers for decades, resulting in numerous theories. Condensational growth rates of droplets to diameters (D) of ~20 µm is well understood, and collision-coalescence is well-evidenced for producing drizzle/rain size drops with those significantly larger than 20 µm (Lamb and Verlinde, 2011). However, for the initiation of collision-coalescence to occur and produce precipitation, drops must first exceed sizes much larger than 20 µm (e.g., Jonas 1996). The range of drop sizes constituting





this "gap" is often termed the "bottleneck" size range, and the production of these drops has remained uncertain for decades.
Proposed mechanisms producing drops within the bottleneck size range include but are not limited to the presence of giant cloud condensation nuclei (e.g., Woodcock 1953; Blyth et al. 2003; Dziekan and Pawlowska 2017; Dziekan et al. 2021), direct long wave cooling of droplets by emission of thermal infrared radiation near cloud top (Zeng, 2018), inhomogeneous mixing via entrainment (Baker et al., 1980) and various mechanisms related to turbulence (e.g., Chandrakar et al. 2024; Pinsky and Khain 2002; Siebert and Shaw 2017; Lu et al. 2018).

Additional growth mechanisms involve drop clustering (i.e., positive spatial correlations between drops) in various manners (e.g., Shaw et al. 1998; Bodenschatz et al. 2010; Madival 2019). Drop clustering is suspected to occur in varying cloud types given sufficiently large Reynolds numbers as evidenced by in situ observations (e.g., Dodson and Small Griswold 2019; Larsen et al. 2018; Dodson and Small Griswold 2022; Bateson and Aliseda 2012; Kostinski and Shaw 2001; Glienke et al. 2020), which is a consequence of processes such as inertial clustering and entrainment-mixing (Beals et al., 2015; La et al., 45 2022). Previous in situ observation studies differ in their conclusions about correlations between drop clustering and drop sizes (Glienke et al., 2020; Marshak et al., 2005; Small and Chuang, 2008; Thiede et al., 2025). These observational studies vary in their usage of instrumentation, spatial scale analysis as well as methods for diagnosing/quantifying the degree of drop clustering. However, to the authors' knowledge all previous studies have related drop sizes to the clustering of drop systems (i.e., sample volumes), and not to the clustering "surrounding" the individual drops being investigated. It is plausible this 50 analytic paradigm fails to relate individual drops to corresponding small scale features (e.g., microscale vortices, supersaturation fields, etc.). Additionally, different biases are associated with different drop clustering metrics and counting statistics of droplets can often be insufficient for obtaining robust clustering diagnoses of individual sample volumes (Baker and Lawson, 2010; Larsen et al., 2018; Shaw et al., 2002). These conditions motivated this study to relate drop sizes directly to the degree of "clustering" around individual drops, focusing on the bottleneck size range. Namely, we will determine the 55 likelihood that drops of a given size will have a significantly high number of drops surrounding them as well as likelihoods that they are significantly isolated from neighboring drops. Section 2 introduces the instrumentation used in this study and discusses the individual drop clustering diagnosis. Section 3 outlines two types of analysis performed in this study: 1) determining the relation of drop size to individual drop clustering and 2) correlating individual drop clustering trends with other parameters. Section 4 provides results and Section 5 includes further discussion of the findings. Section 6 provides 60 concluding remarks.

## 2 Instrumentation and methodology

### 2.1 Instrumentation and field measurements

Data is taken from the National Science Foundation (NSF)/National Center for Atmospheric Research (NCAR) Cloud System Evolution in the Trades (CSET) campaign, which consisted of 16 research flights targeting the stratocumulus to



cumulus transition over the Northeastern Pacific (Albrecht et al., 2019) using the NSF/NCAR G-V aircraft. The standard flight plan called for repeated situ sampling patterns, each of which included level legs below-, in- and above-cloud and sawtooth legs across cloud top.

This study primarily relies on observations from the Holographic Detector for Clouds (HOLODEC), which acquires particle size distribution information for constant sample volumes of ~13 cm$^3$ at a rate of 3.3 Hz (Fugal and Shaw, 2009). For flight
speeds within regions relevant to this study (~130 m s$^{-1}$), this results in samples obtained every ~40 m. The HOLODEC measurements provide drop size distribution information of drops with diameters from about 6–1000 µm, as well as the 3D spatial coordinates of the drops. The drop clustering methodology (described in Section 2.2) is applied to the HOLODEC measurements.

Measurements acquired from other instruments are collocated to HOLODEC measurements, and are all reported at 1 Hz
resolution (~130 m resolution) unless specified otherwise. This includes temperature and water vapor measurements to derive relative humidity (*RH*), which are acquired from the Rosemount temperature probe and Vertical-Cavity Surface Emitting Laser hygrometer (Zondlo et al., 2010). Because of differences in the sampling rates, 3 or 4 holograms are identified with the temperature and *RH* measurements, and this may introduce some uncertainty into the analysis. A W-band HIAPER Cloud Radar (HCR) having a 0.5 Hz resolution is also utilized to identify the lowest altitudes of
condensate/precipitation in the presence of rain/drizzle. This is possible since the HCR is nadir-pointing for in-cloud flight legs. The lowest condensate/precipitation echoes are defined as the lowest range gate from the aircraft having reflectivity exceed -60 dBZ (no lower clouds or fog are considered). This is to explore how HOLODEC measurements correspond with subsiding condensate. Although no efforts are made to distinguish between precipitation and cloud base, the frequency of virga associated with stratocumulus clouds from CSET is predicted to be ~80% (Schwartz et al., 2019).

Although the HOLODEC data were processed for all research flights, there is limited data availability since only a few flight legs had relatively long durations in-cloud (shown below). Observations are also limited to level flight legs due to possible measurement biases and consistency with previous studies (Glienke et al., 2017, 2020; La et al., 2022; Larsen et al., 2018; Larsen and Shaw, 2018). Table 1 shows flight legs from which holograms are taken for the analysis. Data is only taken from flight legs that have at least 100 holograms meeting the required conditions for the analysis (Section 2.2). Note that this
excludes <5% of all available holograms, and does not significantly impact results.




| Flight legs for analysis | | | |
|---|---|---|---|
| **Flight leg** | **Time (UTC)** | **$D_{max} > 25\ \mu m$** | **$D_{max} > 30\ \mu m$** |
| RF02A* | 16:18:40–16:31:40 | 679 | 538 |
| RF02B* | 17:15:00–17:26:40 | 1241 | 537 |
| RF10A* | 16:31:40–16:41:40 | 1487 | 832 |
| RF10B* | 17:20:00–17:28:30 | 977 | 642 |
| RF04 | 17:30:00–17:42:30 | 224 | 126 |
| RF08 | 16:56:40–17:06:40 | 260 | 190 |
| RF11 | 20:18:20–20:29:10 | 150 | 93 |
| RF12A | 21:00:00–21:19:10 | 148 | 141 |
| RF12B | 17:50:00–18:01:40 | 230 | 119 |
| RF13 | 21:15:00–21:22:30 | 139 | 91 |
| RF14 | 17:13:20–17:25:00 | 135 | 93 |
| RF15 | 18:20:00–18:30:00 | 269 | 171 |

**Table 1: List of flight legs used in this paper. Flights including multiple flight legs are listed alphabetically in the order they occurred (shown in Flight leg column). The number of holograms having maximum drop diameters exceeding 25 μm and 30 μm are shown for each flight leg. The first four flight legs within the bolded box and with**
**asterisks are the only ones used in the analysis in Section 4.2. These flight legs are also evaluated for drop clustering in Larsen et al. (2018).**

Due to different in-cloud sampling durations amongst the flight legs, 75% of all available holograms are observed in RF02A,B and RF10A,B (having maximum drop diameter exceeding 25 μm; Table 1). Those flight legs were evaluated in
Larsen et al. (2018), who determined radial distribution functions using HOLODEC measurements separately for each flight leg.

## 2.2 Methodology

Whereas previous studies have quantified droplet clustering of the HOLODEC's respective sample volumes (Glienke et al., 2020; La et al., 2022; Larsen et al., 2018; Larsen and Shaw, 2018; Thiede et al., 2025), results here evaluate drop clustering
on a drop-by-drop basis. In-cloud samples are defined where drop concentrations exceed 100 (per ~3.3 cm$^3$) for consistency with previous clustering analyses of HOLODEC measurements acquired during CSET (La et al., 2022; Larsen et al., 2018;





Larsen and Shaw, 2018), which select this threshold to achieve adequate counting statistics when determining drop clustering. Note that Wood et al. (2018) found approximately half of the clouds sampled during CSET have "ultra clean" drop concentrations (~10 cm⁻³), meaning these clouds are not included in the analysis. This study also adapts a similar

sample volume as these studies, which is smaller than that reported by the HOLODEC. The reported sample volume dimensions of approximately 1 cm × 1 cm × 13 cm are reduced to 0.6 cm × 0.6 cm × 10 cm, since drop spatial coordinates are more reliable within the inner portion of the sample volume (Larsen et al., 2018). An additional inner sample volume, defined as the guard rail and discussed in the following subsection, is applied so drop sizes are only evaluated within the spatial volume of 0.3 cm × 0.3 cm × 9.7 cm. Previous studies restrict their analysis to drops with $D > 10$ μm as they are

assumed to have the best detectability. Drops used here are restricted to sizes of $D > 12$ μm as a lower limit (also for the in-cloud condition), although sensitivity tests using $D > 10$ μm produce results consistent with those shown throughout this study (discussed further in Appendix B).

### 2.2.1 Introduction to the Radial Distribution Function

Multiple methods exist to quantify droplet clustering, some of which are reviewed in Shaw et al. (2002). The radial

distribution function ($g(r)$) determines the number density of particles as a function of distance from a reference particle, and may be used to quantify particle clustering given three-dimensional particle spatial coordinate data is available. It can be expressed as

$$g(r) = \sum_{i=1}^{N} \frac{\psi_i(r)/N}{(N-1)\left(\frac{dV_r}{V}\right)},$$  (1)

where $\psi(r)$ is the number of particles surrounding the $i$th particle within the surrounding spherical shell volume between radii

$r - \Delta r/2$ and $r + \Delta r/2$, $V$ is the measurement volume over the entire hologram, $N$ is the number of drops within the guardrails and $dV_r$ is the measurement volume enclosed within shells having radii $r - \Delta r/2$ and $r + \Delta r/2$. The equation takes advantage of the notion that a Poisson distribution has evenly distributed particles within a sample volume, which is represented in the denominator of equation 1. This means $g(r)$ indicates a greater degree of drop clustering as values increase above 1. An idealized depiction of cloud drops in a hologram is shown in Fig. 1.




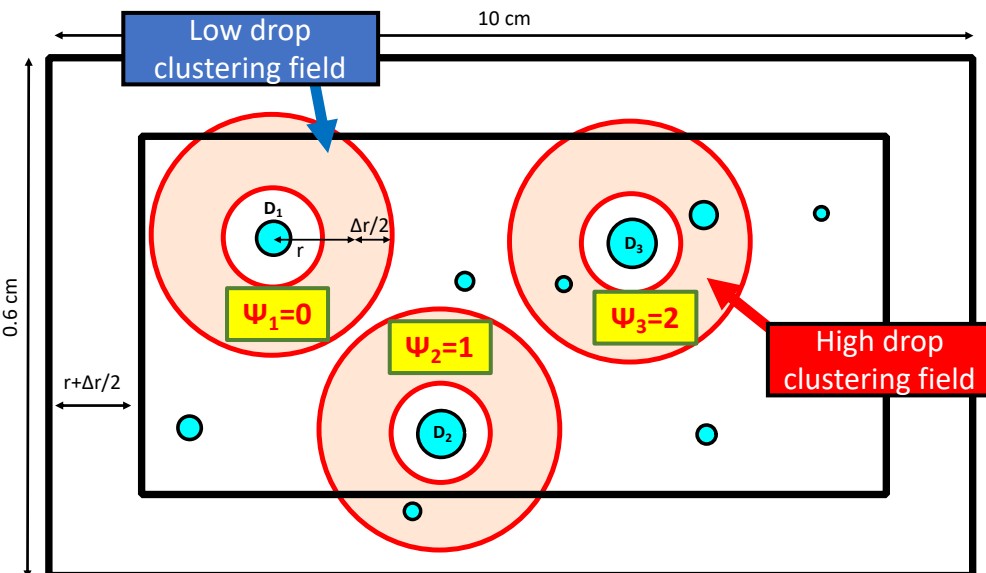

**Figure 1: An idealized depiction of drops shown with different numbers of neighboring drops in a hologram. Low (high) drop clustering fields (DCFs) correspond to drops with a significantly low (high) number of neighboring drops (how significance is defined is described in the text). $V$ is the volume of the innermost box and $dV$ is the volume within the light-red shaded spherical annulus volumes of the respective droplets (variables from Eq. 1). Distances are not to scale.**

This study employs a guard rail approach which only computes $g(r)$ among the $i$th particles that are within a certain distance from the measurement volume edges in order to prevent counting biases when computing pairwise correlations. The guard rail distance is the maximum possible shell radius used when computing the radial distribution function ($r + \Delta r/2$ in Fig. 1). A detailed overview and visualization of the radial distribution function and guard rail approach can be found in Larsen and Shaw (2018). Two labels within Fig. 1 are denoted "high drop clustering fields" and "low drop clustering fields" (DCFs), which correspond with the drops having the most and fewest neighboring drops within their respective shells. How high and low DCFs are diagnosed using multiple shells sizes is discussed below.

Rather than quantify the clustering of a system (i.e, sample volume) of droplets, this study directly relates drop sizes to their respective DCFs. The proposed DCF methodology will conceptually mirror DCFs used for determining radial distribution functions, although clustering fields will not be computed at incremental distances away from the droplet centroids. Rather, DCFs will incrementally increase in size while maintaining a constant lower bound. The following section will detail the methodology including how droplets are diagnosed as high/low DCF.





### 2.2.2 Particle-by-Particle Methodology

This study aims to directly compare how occurrence frequencies of high and low DCFs differ for different drop sizes. The methodology used to classify high and low DCFs is introduced below, which is derived from the radial distribution function to determine clustering statistics around individual drops. If one were to determine $g(r)$ for an individual drop ($g_i$), the radial distribution function in Eq. 1 simplifies to

$$g_i(r) = \frac{\psi_i(r)}{(N-1)\left(\frac{dVr}{V}\right)}, \tag{2}$$

where the summation term is no longer included. Note that total drop concentrations are still included in the denominator to weight the observed number of neighboring drops by the number of drops following a Poisson distribution. However, $g_i$ of individual drops are heavily biased by the total drop concentration term ($g_i$ is greatest at the lowest drop concentrations; Supplementary Fig. S1). This is because low, non-zero $\psi$ will significantly exceed the "expected" number produced by the denominator term (a form of counting statistic error). Further, one can intuit that greater drop concentrations will increase likelihoods of drops neighboring each other at closer distances relative to lower drop concentrations (given constant sample volumes).

In order to compare DCFs of holograms having notably different drop concentrations, we wish to remove the "expected" (i.e., Poisson) drop count term from consideration. Since we want to determine $g_i$ relative to other drops within their respective holograms, $N$ and $V$ can be removed since they are both constant for each respective hologram. As previously mentioned, DCFs of individual drops are not computed as functions of incremental distance from the drop centroids, but rather as a function of incrementally increasing shell size with a constant lower boundary. This updated individual drop clustering term ($g_{drop}$) is expressed as

$$g_{drop}(r_{min} + \delta r_n) = \frac{\psi(r_{min} + \delta r_n)}{dV_{r_{min} + \delta r_n}}, \tag{3}$$

where $r_{min}$ is a constant lower boundary and $\delta r_n$ is the width of the spherical annuli (i.e., shell), each starting at $r_{min}$ whose width increases with increasing $n$ from $n = 1, \ldots, 7$ (top of Fig. 2; discussed below). Note that $\delta r$ is applied differently than $\Delta r$ in Eqs. 1&2 (illustrated below). It is crucial to maintain constant shell sizes regardless of drop size, rather than vary $r_{min}$ according to respective drop diameters. Having $r_{min}$ equal the droplet radius would result in small drops having larger shells compared with large drops given constant $\delta r$. A constant $r_{min}$ of 50 µm is used here, since it is the maximum distance at which two drops having $D$=50 µm can neighbor each other. Note that the number of drops occurring within $r_{min}$ and all drop surfaces amounts to ~0.00002% of all drops, so results are not impacted by this choice of $r_{min}$. While this $r_{min}$ value means that the DCF may not be well-defined for drops with $D$>50 µm, only ~200 such drops exist in the dataset as defined below, and they are not considered when determining statistical likelihoods of drop sizes for given DCF ranges.

Figure 2 shows an idealized depiction of DCFs within a hologram having multiple shells, where shell sizes increase with a constant lower boundary ($r_{min}$) following $g_{drop}$ (eq. 3). This is seen when comparing $\delta r_1$ and $\delta r_2$ in Fig. 2.





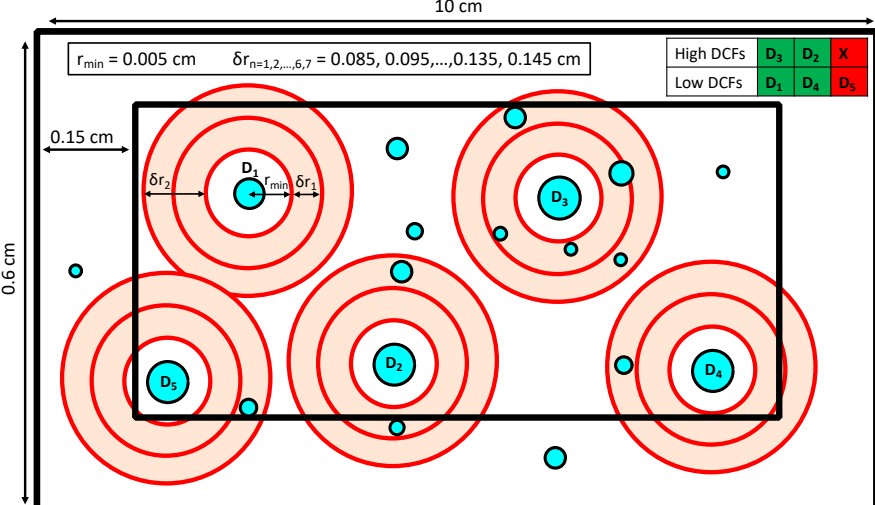

**Figure 2: Idealized depiction of high and low drop clustering fields (DCFs) using multiple shells as determined by $g_{drop}$. DCFs are only shown for select drops ($D_1$–$D_5$). The usage of shells follows the proposed methodology and not following RDF (i.e., the lower bound $r$ [$r_{min}$] stays constant). The upper left lists the concentric shell size range of radii from 0.09–0.15 cm, with each consecutive shell size increasing in distance from $r_{min}$ by 0.01 cm. Note that only two shell sizes are shown for simplicity, although the proposed methodology includes seven concentric shell sizes. The upper right panel denotes how high and low DCFs are sorted for their respective maximum to minimum values. Note that $D_4$ and $D_5$ have the same values of $g_{drop}$, and are therefore randomly sorted (discussed in text). The red shading denotes drops not included in the analysis, since the same number of high DCFs and low DCFs must be available from each hologram (discussed in text). Distances are not to scale.**

The different shell sizes allow for computation of varying "degrees" of high and low DCFs, where shell sizes are integer multiples of $r_{min} + \delta r_1$ (top row of Fig. 2). The lowest DCF values are assigned to drops that do not have any neighbors within the largest possible shell size (i.e., $r + \delta r_n$). Drops diagnosed as having "low DCFs" are then sorted by the largest shell size that does not contain any neighbors for each hologram. Drops diagnosed as having "low DCFs" are then sorted by the largest shell size that does not contain any neighbors for each hologram. This is depicted in Fig. 2 where $D_1$ has the lowest DCF, since it is the only drop with no neighboring drops in all available shell sizes. $D_4$ and $D_5$ have identical clustering fields, with no drops in the smaller shells. They are then randomly sorted following $D_1$, due to their identical clustering fields. To further discretize low DCFs, when multiple drops have no neighboring drops within their respective shells of the same size, they are sorted by the number of drops within the maximum shell size of each drop (i.e., $r_{min} + \delta r = 0.15$ cm). For example, assume there are drops $D_a$ and $D_b$ and both have no neighboring drops within the 0.13 cm shell (i.e., 0.005–0.13 cm from the droplet centroid). $D_a$ and $D_b$ have one drop and three drops within the 0.15 cm shell (i.e., 0.005–0.15 cm from the droplet centroid), respectively. Therefore, $D_a$ is sorted before $D_b$ and has the lower DCF. This means low DCF are primarily




sorted by the largest shell size in which no neighboring drops exist, and when shell sizes are the same, they are sorted by the number of neighboring drops within the maximum shell size. These conditions as well as others (discussed below) are
detailed in Table 2.

| Drop clustering field (DCF) Classification | | | | |
|---|---|---|---|---|
| **Step** | **Primary conditions** | | **Secondary conditions** | |
| 1. Determine in-cloud holograms | $N_{outer}$ > 100 per hologram (28 cm$^{-3}$) | | - | |
| 2. Determine individual drop clustering | Determine $g_{drop}$ for all drops having 12<$D$<50 µm for all shell sizes (i.e., r+δr). | | - | |
| - | **High DCFs** | **Low DCFs** | **High DCFs** | **Low DCFs** |
| 3. Determine high and low drop clustering fields (DCFs) | Select maximum $g_{drop}$ amongst all shell sizes meeting '$N$ for $\psi$' condition below | $\psi$ = 0 for maximum shell size, meeting '$N$ for shell size' condition below | Drops having $D$>50 µm are not included in $\psi$ | Shell size not considered if drop with D>50 µm is within shell |
| | **Additional conditions categorizing high DCFs (top two rows) and low DCFs (bottom two rows)** | | | |
| | **N for $\psi$**    **N<110** | **N: 110-175** | **N: 175-230** | **N: 230-280**   **N>280** |
| | Minimum $\psi$ for high DCFs   $\psi=2$ | $\psi=3$ | $\psi=4$ | $\psi=5$    $\psi=6$ |
| | **N for shell size**   **N<85** | **N: 85-140** | **N: 140-180** | **N: 180-300**   **N>300** |
| | Minimum shell size for low DCFs   $r_{min}+\delta r=0.12\ cm$ | $r_{min}+\delta r=0.11\ cm$ | $r_{min}+\delta r=0.10\ cm$ | $r_{min}+\delta r=0.09\ cm$   $r_{min}+\delta r=0.09\ cm$ |
| 4. Sort high and low DCFs separately | Sort by maximum $g_{drop}$ from greatest to lowest values | Sort by shell size from maximum to minimum shell sizes | - | If shell sizes are identical, sort drop by $\psi(r_{min}+r_{max})$, where $\psi(r_{min}+r_{max})$ = 0.15 cm |
| 5. Final data filtering | Trim sorted array having a greater number of elements (i.e., drops) such that $N_{high\_DCF} = N_{low\_DCF}$ | | Only use holograms with more than 3 DCF pairs (i.e., $N_{high\_DCF}$ > 3 and $N_{low\_DCF}$ > 3) | |
| 6. Monte Carlo – DCF methodology | Shuffle drop sizes amongst drops within the inner guardrails for each hologram 1000 times (i.e., simulations). This creates a counterfactual of no DCF – drop size relationship. Then compare actual DCF – drop size relationships with simulations. | | | |
| Variables | $N_{outer}$: Number of drops in 0.6x0.6x10 cm sample volume <br> $N$: Number of drops within guardrail sample volume (0.3x0.3x9.7 cm) <br> $\psi$: Number of drops within a shell having $r = r_{min}+\delta r$ <br> $D$: Drop diameter <br> $g_{drop}$: Equation 3 | | $N_{high\_DCF}$: Number of drops with high DCFs <br> $N_{low\_DCF}$: Number of drops with low DCFs <br> $r_{min}$: 0.005 cm from drop centroid <br> $\delta r_{max}$: 0.145 cm from $r_{min}$ <br> $\delta r_{n=1,2,...,6,7}$: 0.085, 0.095,...,0.135, 0.145 cm | |

**Table 2: All conditions of the Monte Carlo drop clustering field (DCF) classification method which classifies drops as having either high DCFs or low DCFs. The thickest bolded conditions are associated with Step 3. The middle thick bolded box surrounds conditions associated with the DCF classification of individual drops (Steps 3&4). Both**



**primary and secondary conditions must be met for data to be included in the analysis. The final step references the**
**DCF – drop size Monte Carlo analysis, discussed in Section 3.1. Variable names are provided in the lowest row.**

Determining high DCFs requires more nuanced consideration, since the shell size and the number of neighboring drops must be considered. Since both terms are considered in $g_{drop}$, the maximum $g_{drop}$ is selected among all shell sizes when diagnosing high DCFs. Supplementary Figure S2 shows how different $g_{drop}$ rank amongst each other for varying $\psi(r_{min} + \delta r)$ and shell sizes.

Utilizing larger shell sizes allows for the ability to diagnose increasingly isolated drops. However, larger shell sizes consequently decrease the guardrail distances, which limits the available sample volume (depicted as $r + \delta r$ in Fig. 2). The upper bound of 0.15 cm is therefore chosen to both ideally diagnose isolated drops while producing a sufficient sample of drops within the bottleneck size range.

At this point, the high and low DCF conditions are still insufficient to completely separate drops into the two categories. For
example, $D_4$ and $D_5$ in Fig. 2 only have one drop in the larger shell and no drops in the smaller shell. By the current definitions, these drops would meet the requirements to be categorized as both low and high clustering drops. To ensure drops are appropriately categorized, an additional set of conditions is shown in the tan shading of Table 2. First: for a given $N$, a minimum number of neighboring drops within the largest shell is required to be considered a high DCF (top two rows). The rationale is that a crude consideration of counting statistics can be used to help discretize the drop clustering categories
by applying this consideration towards high DCFs. This is also useful since it prevents a single neighboring drop within a relatively small shell to be potentially selected as the maximum $g_{drop}$.

The second set of conditions requires a sufficiently large shell size for classifying low DCFs (bottom two rows of the tan shaded region of Table 2). Namely, the definition of a low DCF drop changes with the total drop population $N$ within a sample volume, such that the size of the shell that contains no neighboring drops is required to be larger as $N$ decreases as
shown in Table 2. One can intuit that lower drop concentration environments will produce greater likelihoods of isolated drops at the smallest shell sizes compared to high drop concentration environments, warranting this condition. Even applying these two sets of conditions does not exhaustively discretize high and low DCFs categories for all possible combinations of shell size and neighboring drops. Therefore, those drops which meet both conditions are designated as high DCFs. Note again that sorting the DCFs (top right panel in Fig. 2) allows for the ability to focus on the highest and lowest DCFs. Results
will primarily focus on the highest and lowest DCFs, which avoids considering these "problematic" DCFs (i.e., those meeting both conditions). DCFs not meeting the conditions in Tables 2&3 are categorized as "Neither".

In order to perform drop-by-drop comparisons, the same number of high and low DCFs are selected from each hologram to control for environmental conditions. Namely, by selecting the same number of high and low DCFs from each hologram, we





control for environmental factors and bulk properties (e.g., *RH*, *N*, etc.) since the same number of high and low DCFs will be
associated with any given environmental/microphysical variable. To achieve this, the DCF category with the greater number
of drops is reduced (randomly) to have the same number of drops as the lower category (Step 5 in Table 2). Sampling with
replacement of the smaller drop category in order to retain a greater number of droplets is also possible. However, the
current methodology results in approximately one third of drops having high DCFs, low DCFs and DCFs within these two
ranges (shown below). And since results will primarily focus on the highest and lowest DCFs (Section 3), there is no need
for sampling with replacement. The requirement of at least 4 high and low DCFs within each hologram (secondary condition
of Step 5 in Table 2) will be discussed in relation to the Monte-Carlo – DCF methodology (Step 6) below, and results in ~5%
of available holograms being discarded.

The conditions used to classify high and low DCFs are chosen in order to separate drops into terciles of those having high
DCFs, low DCFs and DCFs meeting neither condition. This "tercile approximation"; is shown in Fig. 3B, which shows *N* in
the x-axis and drop concentrations associated with high DCFs (red points), low DCFs (blue points) and neither (green points)
for their respective holograms ($N_{DCF\_categories}$). The one-to-one line and the one-to-three line are shown by the dotted and
dashed lines, respectively. Since the same number of drops from high and low DCFs are selected from each hologram, the
lower value of these two (i.e., the number of drops used in the analysis) is shown by the black points. The drop
concentrations used in the analysis approximately follow the one-to-three line, signifying that terciles are approximately
captured among DCFs for all the holograms.





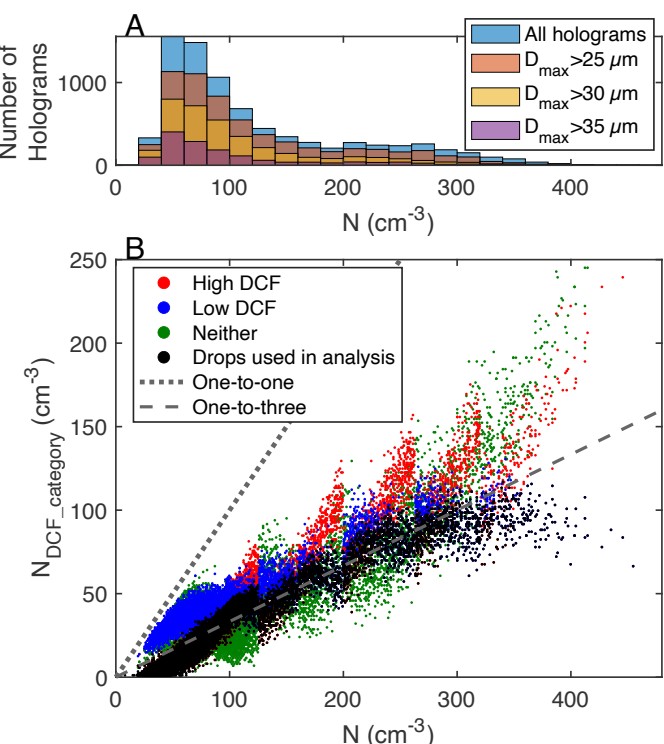

**Figure 3: A) Histogram of available holograms throughout the campaign. Different distributions account for holograms with different maximum drop diameters ($D_{max}$). B) The number of high DCFs, low DCFs and DCFs meeting neither category ($N_{DCF\_category}$) related to the holograms' drop concentrations ($N$). Note that $N$ refers to drop concentrations with $D>12$ μm.**

Figure 3A shows the total number of holograms used in this study (blue bars). Since the focus of this study is on drops in the bottle-neck drop size range, the total number of holograms with maximum drop diameters ($D_{max}$)>25 μm, $D_{max}$>30 μm and $D_{max}$>35 μm are also provided to illustrate the number of available holograms used in the following analysis. Only holograms with $D_{max}$>25 μm will be analyzed in this study.

## 3 Statistical testing for drop clustering

Up to this point, a methodology has been introduced to classify high DCFs (i.e., those with a high number of neighboring drops) and low DCFs (i.e., drops notably isolated from surrounding drops). DCFs are categorized relative to the drop concentrations of their respective holograms, meaning high DCFs will likely contain fewer drops in holograms with low drop concentrations compared with high DCFs in holograms having high drop concentrations. This section introduces two




methodologies to evaluate the relationships of drop sizes to their respective DCFs—both of which rely on some form of Monte Carlo analysis. Section 3.1 is a description of how the statistical likelihood of DCFs are determined in relation to their respective drop sizes. Section 3.2 introduces a methodology to discern which environmental/microphysical parameters are most notably related to DCF signals of interest.

## 3.1 Methodology #1: Monte Carlo-DCF methodology

To determine how high and low DCFs relate to drop sizes (i.e., whether clustering tends to be greater around larger and/or smaller drops), a statistical methodology using Monte Carlo simulations is applied to each hologram. This will be referred to as the Monte Carlo-DCF methodology, and is shown in the final step of Table 2. Namely, for each simulation the drop sizes within the inner guardrail sample volume are randomly shuffled amongst the drops within this sample volume. Only the drop sizes are randomly sorted while the original drop spatial coordinates remain unchanged — meaning the DCFs will be identical for each hologram. This is to provide a counterfactual where no relationship between DCFs and drop sizes exists. The same number of drops meeting the high/low DCF classification are then randomly selected from each hologram. DCFs of the actual drop sizes are then related to those of the randomly assigned drop sizes from the Monte Carlo simulations. One thousand Monte Carlo simulations are run for all holograms used in this study. Findings directly applying this methodology are provided in Section 4.1.

## 3.2 Methodology #2: Monte Carlo-Hologram Comparison methodology

The drop clustering method above allows us to examine the relationship between drop size and its DCF as a function of some environmental variable (e.g., whether the holograms come from subsaturated or supersaturated regions). However, this requires the manual selection of holograms when exploring DCF–drop size relationships and may result in Type-I errors (illustrated in Section 4). A second methodology is proposed here to discern which variables are most notably associated with correlations between DCFs and drop sizes, without the need to manually select specific holograms. This is called the Monte Carlo-Hologram Comparison methodology.

For each simulation using this Monte Carlo-Hologram Comparison methodology, a specified number of holograms are randomly selected from the set of available holograms. DCF–drop size relationships are then determined for this set of holograms using the Monte Carlo-DCF methodology from Section 3.1. Simulations are separated into one of two categories: those where an "isolated large drop trend" (i.e., all drops within bottleneck range are most likely to be isolated from other drops) is observed and those where it is not. In other words, the former category contains simulations having high likelihoods of isolated large drops (HILD) and the latter category contains simulations not possessing this signal (OTHER). The associated environmental/bulk microphysical variables are also recorded from the simulation sets of the two categories, and are then compared with each other to determine likelihoods that these variables are associated with this isolated large drop trend. Namely, normalized frequency distributions of the given variable are produced for each of the two categories,



and the difference between the distributions is computed over each size bin. This methodology is outlined in Table 3, and a step-by-step description of the methodology is provided in Appendix A.

| Monte Carlo – Hologram Comparison methodology | | | |
|---|---|---|---|
| **Step** | **Details** | | |
| 1. Randomly sample X holograms from available holograms | Compute 20,000 Monte Carlo simulations | | |
| 2. Run sets of simulations using different X (i.e., batch sizes) | Batch sizes = 100, 200, 300,… , 2200 (each set is 20,000 simulations) | | |
| - | Categories | HILD | OTHER |
| 3. Classify simulations as those with a high likelihood of isolated large drops (HILD) or those not capturing this trend (OTHER) | Percentiles (all conditions must be met to be included in a category) | D: 25–30 μm > 95th D: 30–37.5 μm > 95th D: 37.5–50 μm > 90th | D: 25–30 μm < 80th D: 30–37.5 μm < 80th D: 30–37.5 μm < 80th |
| 4. Remove batch sizes having low statistical power | For a batch size to be considered, there must be at least 100 HILD and 100 OTHER simulations available | | |
| 5. Produce normalized frequency distributions for each set of simulations | Produce distributions separately for HILD and OTHER simulations for each batch size | | |
| 6. Compute the difference of HILD and OTHER distributions for each batch size | - | | |

**Table 3: Components of the Monte Carlo-Hologram methodology. Details/further clarification of the methodology are described in the text and in Appendix A.**

Only holograms from flight legs RF02A,B and RF10A,B are included in the Monte Carlo-Hologram Comparison

methodology, because the random selection of holograms will consequently draw from these flight legs having significantly more holograms. This is also done in order to perform separate flight leg comparisons, which will become evident in Section 4.2. Note that each individual simulation above may include holograms from multiple flight legs. A more detailed description of the Monte Carlo-Hologram Comparison methodology is provided in Section 4.2 and Appendix A.

**4 Results**

**4.1 Initial DCF – drop size relationships**

Figure 4 shows results from the Monte Carlo-DCF methodology for different sampling conditions overlying the respective panels (bold text). Results hereafter are restricted to holograms having maximum drop diameters $D_{max}$>25 μm. The overlying elongated panels (A–D) show the percentiles of actual drop sizes having either high or low DCFs (red and blue markers,



respectively) in relation to those of the Monte Carlo simulations. These panels show whether drops of a certain size are more/less likely to be associated with a given DCF category, where higher (lower) percentiles are associated with drop sizes being more (less) likely to be associated with either high or low DCFs. Specifically, the actual occurrence frequencies of drops having high and low DCFs within the respective drop size bin ranges (lightly shaded dotted lines) are directly related to occurrence frequencies from the Monte Carlo simulations. These occurrence frequencies are shown in Fig. 4A1–D1, where the thick red and blue lines are the actual occurrence frequencies of drops having high and low DCFs, respectively. The Monte Carlo occurrence frequencies are shown as thin red and blue lines, although they are often overlapped by the actual DCF distributions.

To help clarify the findings, we first focus on the top left panels: Figure 4A shows percentile results for all drops categorized as high and low DCFs amongst all holograms used in this study. A few "statistically significant" drop size ranges can be found, i.e., DCFs either below or exceeding the 5th and 95th percentiles, respectively (represented by the diamonds and stars). However, all of the drop sizes exceeding 25 µm are associated with relatively high percentiles of low DCFs, including a statistically significant drop size range from 30–37.5 µm. To better illustrate the comparison between the actual and Monte Carlo distributions, the 30–37.5 µm drop size bin from Fig. A1 is magnified and displayed in the middle of the figure. The Monte Carlo distributions can now be seen as the thin red and blue lines (although giving a partial appearance of purple lines due to their overlap), corresponding to the high and low DCFs, respectively. The actual low DCF frequency (thick blue line) exceeds most of the Monte Carlo frequencies (thin blue lines), specifically exceeding over 95% of them. This means drops within this size range are significantly likely to have a low DCF, i.e., to be isolated from the nearest drops.

Section 2 discussed how drops with varying degrees of high and low DCFs are sorted. And since the same number of high and low DCFs are selected from each hologram, percentiles of DCF "degrees" can be obtained for direct comparison with each other. To focus on drops with increasingly higher DCFs (i.e., larger maximum values of $g_{drop}$) and lower DCFs (drops with greater distances from the nearest drops), Fig. 4B shows results restricted to DCFs above the 50th percentile, and Fig. 4D shows results in the upper tercile (DCFs>67th percentile). To help understand the degrees of high/low DCFs and how they are selected from the sorted DCF values, the results in Fig. 4A,A1 correspond with analyzing drops $D_{1–4}$ from the hologram in Fig. 2 (green shaded drops in top right panel), and results in Fig. 4B,B1 correspond with only analyzing drops $D_{1\&3}$.

Results from panel A are considered our exploratory analysis by looking at all available data. Excluding the lowest drop size bin, a trend of increasing percentiles with increasing drop size is observed for low DCFs (blue markers). Because of this, we hypothesize large drops ($D$>25 µm) should be associated with the lowest DCFs, likely because dry air preferentially evaporates small droplets due to their greater surface area to volume ratio relative to larger drops. We test this hypothesis by looking at drops with increasingly lower DCFs (Panels B and D). Consistent with the hypothesis, DCF percentiles increase for large drops moving from panels A to B and finally D—as seen by the increasing number of blue diamonds and stars. Namely, percentiles of large drops having low DCFs increases with lower DCFs. In other words, *drops within the bottleneck size range are most likely to be significantly isolated from surrounding drops*. This finding is further suggested by results





with the moderate DCF values below the 50$^{th}$ percentiles (Fig. 4C), which show no significant differences of low DCFs

between the observations and Monte Carlo simulations for large drops (i.e., no increased likelihoods of large drops having low DCFs, seen by blue points for $D>25$ µm).

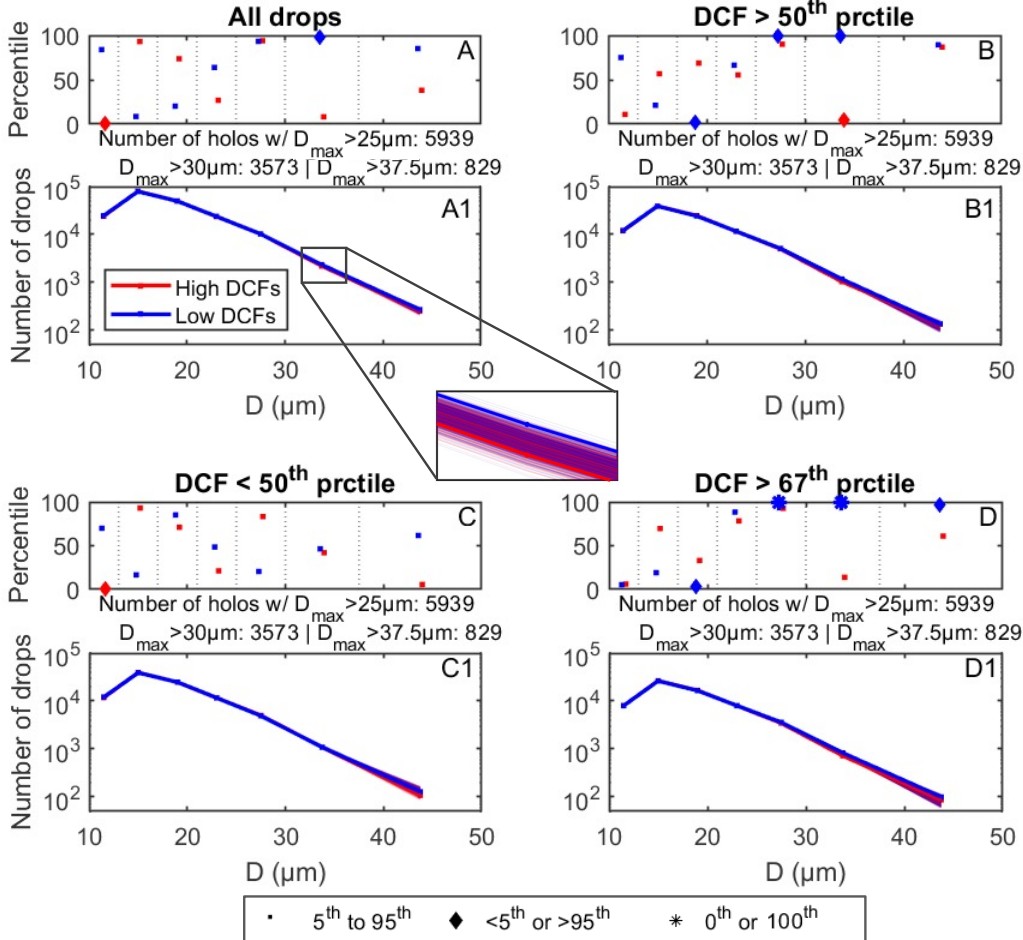

**Figure 4: A) Percentiles of drop occurrence frequencies in relation to Monte Carlo–DCF simulations for all drops from all holograms used in the analysis – described in further detail in the text. Red markers correspond to drops with high DCFs and blue markers correspond to drops with low DCFs. Similar results are shown but restricted to higher and lower DCFs than the respective holograms' median values (i.e., degrees of high and low DCFs > 50$^{th}$ percentiles); B), below the median values (C) and in the upper tercile (D). Markers are denoted by multiple marker**

**shapes, where points correspond to percentiles ranging from the 5$^{th}$–95$^{th}$, diamonds correspond to percentiles below (exceeding) the 5$^{th}$ (95$^{th}$) percentiles, and stars correspond to percentiles below (exceeding) the lowest (highest) simulated frequencies (bottom legend). The number of drops in each DCF category is shown underlying the**



**respective percentile plots (thick lines; A1–D1), which also include thin red and blue lines denoting results from the Monte Carlo simulations. Note that Monte Carlo simulation frequencies nearly overlap those of the observed DCF**

**categories. Results are restricted to holograms having $D_{max}$ >25. The number of holograms having different $D_{max}$ are shown immediately below the respective percentile panels.**

Results in Fig. 4 were taken from all available holograms, meaning drops were selected from all holograms regardless of environmental or other external conditions. This is why the number of holograms remains unchanged underlying each pair of

panels. Figure 5A,C shows results separated into relatively large regions (~100 m) of subsaturated and supersaturated conditions (*RH*<99.5% and *RH*>100.5%, respectively) and Fig. 5B,D shows results separated by relatively low and high drop concentrations. Results here are limited to DCF percentiles in the upper tercile (similar to Fig. 4D). Findings show that the increased likelihoods of large drops being isolated from neighboring drops are observed for subsaturated conditions and relatively low drop concentrations (low DCF exceed the 95$^{th}$ percentile for *D*>25 μm in 5A,B). Both of these conditions are

expected with entrainment, as environmental dry air engulfed into the cloud commonly decreases drop concentrations (via inhomogeneous mixing). However, most samples from CSET have these relatively low drop concentrations, which are expected in marine environments (including the CSET measurements: Wood et al. 2018; Bretherton et al. 2019). The vast majority of high drop concentrations come from flight legs #3 and #4 (both with median *N* exceeding 115 cm$^{-3}$, not shown), which will be discussed further below.






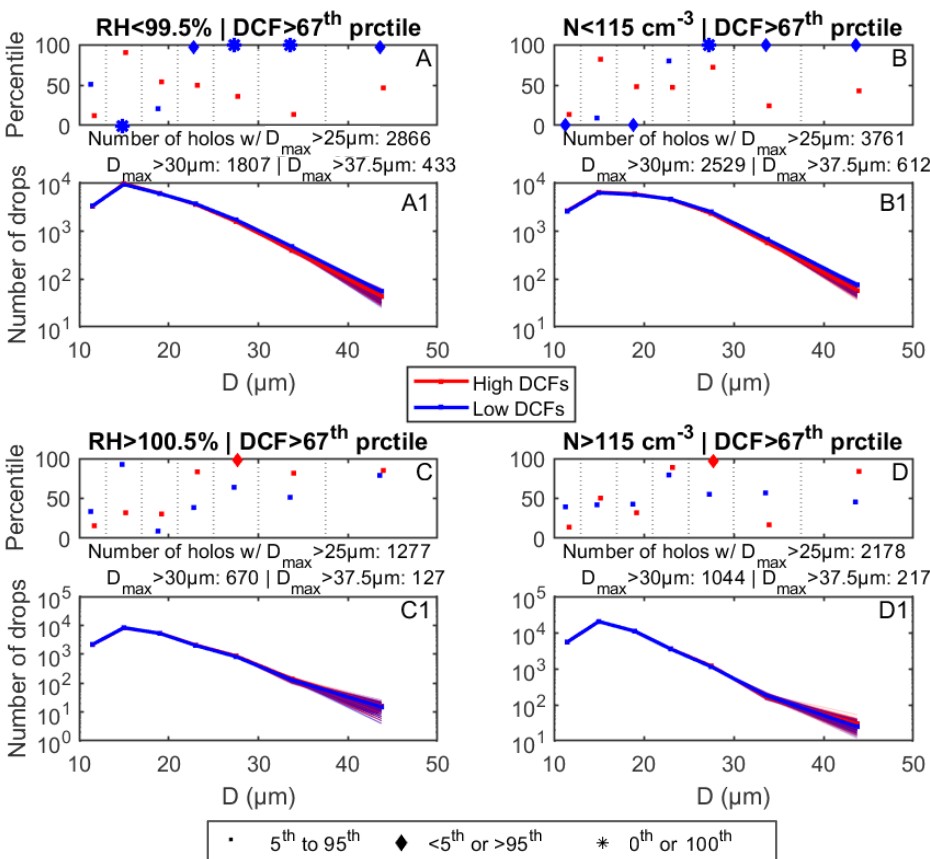

**Figure 5: Similar to Fig. 4, except holograms are separated into subsaturated conditions ($RH$<99.5%; A), low drop concentrations ($N$<115 cm$^{-3}$; B), supersaturated conditions ($RH$>100.5%; C) and high drop concentrations ($N$>115 cm$^{-3}$; D).**


Due to the limited number of flight legs available for analysis, an analysis of DCFs for separate flight legs is warranted. This is to determine whether macro-scale features not evident from localized in situ measurements may correspond to this isolated large drop trend. Two examples are a dependence on cloud regime or background aerosol conditions. Aerosol properties cannot be determined locally due to major biases associated with in-cloud measurements from instrumentation such as
aerosol spectrometers and cloud condensation nuclei counters (e.g., Hudson and Frisbie 1991). As previously mentioned, 75% of available holograms having $D_{max}$>25 μm are from flight legs RF02A,B and RF10A,B (Table 1). Figure 6 focuses on these flight legs—showing percentile data similar to Fig. 4&5 but for each of these flight legs separately.





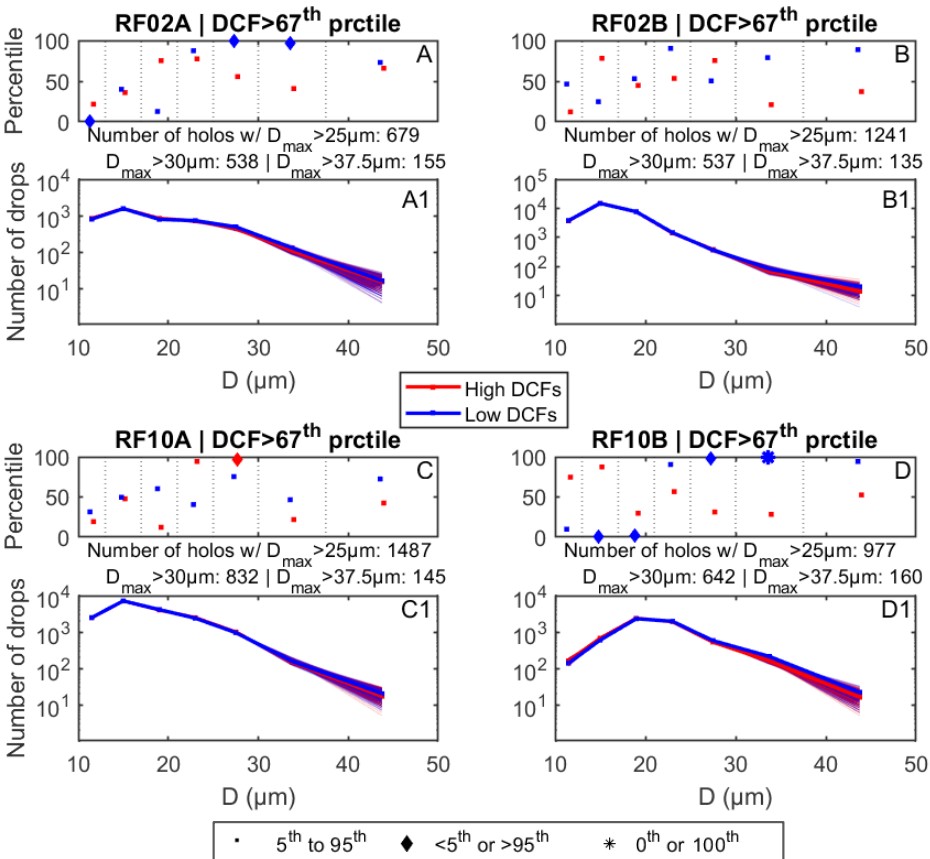

**Figure 6: Similar to Fig. 4&5 but shown for flight legs RF02A,B and RF10A,B (A&A1, B&B1, C&C1 and D&D1, respectively).**

Amongst the four flight legs, the isolated large drop trend is only observed for RF02A and RF10B (Fig. 6A,D). This will be motivation to focus attention on DCF trends for these two flight legs in the following section.

**4.2 Monte Carlo simulations – Hologram comparisons**

Manually selecting holograms (e.g., separating results by holograms in sub-/supersaturated environments) may not be the best way to determine which variables are associated with the isolated large drop trend. Further, this approach may result in a particularly egregious case of p-hacking (i.e., testing combinations until "satisfactory" results are obtained; possibly resulting in Type-I errors). To mitigate this, we employ a different Monte Carlo analysis where each simulation randomly selects a number of holograms and determines the DCF–drop size likelihoods as in Fig. 4–6.

Multiple simulation sets are run to perform comparisons of holograms similar to those in Fig. 4&5, which is detailed in Table 3. Each simulation set selects a different number of holograms (i.e., batch size) for the simulations. A variety of batch



sizes ranging from 100 to 2200 are used since it is unclear which/how many samples are the main contributors to the isolated large drop trend. Results are separated into two categories: (1) High likelihood of Isolated Large Drops (HILD) and (2) those

with lower likelihoods (OTHER). The drop bins used to determine likelihoods are kept consistent with those in Fig. 4–6, and the conditions of each category are shown in Table 3. Note the less stringent threshold applied to the 37.5–50 µm diameter bin (exceeding the 90$^{th}$ percentile rather the 95$^{th}$ as for the 25–30 µm and 30–37.5 µm bins) approximately doubles the available number of simulations in the HILD category. This "lax" condition we argue is applicable due to the relatively low number of drops in this largest $D$ bin. In order to improve the empirical power of the sample sets (i.e., number of simulations

rejecting the null hypothesis divided by the total number of simulations), simulation sets must have at least 100 HILD simulations and 100 OTHER simulations to be included in the analysis. Normalized frequency distributions of relevant parameters are then produced for each simulation set. These distributions are separately produced for HILD and OTHER simulations, and the difference of these two distributions is computed over all respective bins. Distributions are normalized since the Monte Carlo sampling captures different numbers of HILD and OTHER simulations for each simulation set (Seen

in Fig. A1A in Appendix A). Further description and illustration of the methodology is provided in Appendix A.

Figure 7 shows the difference in the normalized frequency distributions (i.e., HILD – OTHER) for multiple parameters as box plots of all available simulation sets. Results are separately shown for the analysis applied to all holograms having $D_{max}$ > 25 µm (red box plots) and for $D_{max}$ > 30 µm (blue box plots) to test for consistency. Each simulation set has a separate

batch size, and there are 17 simulation sets for $D_{max}$ > 25 µm and 10 simulation sets for $D_{max}$ > 30 µm (discussed in Appendix A). The gray lines are histograms showing the number of holograms with $D_{max}$ > 25 µm, which correspond to the right ordinates. Note that a cubic interpolation is applied to the histograms, which have the same bin sizes as the overlying box plots. The HILD – OTHER differences show how given parameters are related to the isolated large drop trend. For example, positive values of HILD – OTHER show that a greater number of holograms possessing values of a given parameter are

associated with the isolated large drop trend, and vice versa. We reminder the reader that results here are only shown for flight legs RF02A,B and RF10A,B. Supplementary Figure S3 is similar to Fig. 7 except all twelve flight legs are used, and all major trends discussed below are still observed.





**Figure 7: Differences in the normalized frequency distributions of HILD and OTHER for number of flight leg samples (A), *RH* (B), *N* (C), number weighted mean diameter (*mean D*) (D), the standard deviation of drop diameter (*$\sigma_D$*) (E) and *$D_{max}$* (F). Results are displayed as box plots containing differences for all simulation sets (described in the text and Appendix A). Red and blue box plots display simulation set differences selecting from holograms with *$D_{max}$*>25 µm and *$D_{max}$*>30 µm, respectively. The dashed line is shown where the difference equals 0. Colored shading overlying and underlying the dashed line corresponds to upper and lower bound confidence intervals, respectively. Confidence intervals are displayed as the maximum and minimum 80th percentiles of statistical significance taken from the 25th, 50th and 75th percentile differences for each respective bin (i.e., corresponding to simulations sets of the bottom, middle dot and top of the box plots, respectively). A description of how significance is determined as well as justification for selecting an 80th percentile level of significance is provided in Appendix A. The grey lines show histograms of the respective variables from the dataset corresponding to the right ordinates. Histograms have the same bin sizes as the overlying box plots and have a cubic interpolation applied to them.**



The HILD – OTHER differences for number of samples within flight legs, $RH$ and $N$ (Fig. 7A,B,C) are consistent with trends from Fig. 5&6. The HILD simulations have a greater number of samples taken from flight legs RF02A and RF10B (Fig. 7A), consistent with flight legs capturing the isolated large drop trend in Fig. 6. The HILD simulations also have a greater number of subsaturated samples as well as a greater number of relatively low drop concentrations ($N$<115 cm$^{-3}$), consistent with trends in Fig. 5. Trends are broadly consistent between those selecting holograms from different $D_{max}$ datasets (red and blue box plots).

The HILD – OTHER differences are shown for the number weighted mean diameter (*mean D*) in Fig. 7D. The HILD simulations are primarily associated with the largest *mean D* (>23 µm), which is consistent with our theory of smaller droplets preferentially evaporating due to their greater surface area to volume ratio. To explore whether there is any evidence of entrainment-mixing drop size distribution broadening, HILD – OTHER differences are similarly shown for the standard deviation of drop diameters ($\sigma_D$) (Fig. 7E) and for $D_{max}$ (Fig. 7F). Positive differences are associated with the largest $\sigma_D$ ($\sigma_D$>5 µm), consistent with entrainment-mixing drop size distribution broadening. Finally, positive differences are observed for relatively large $D_{max}$, ranging from 35–45 µm. However, holograms with $D_{max}$ exceeding 45 µm are not associated with the isolated drop trend. This either suggests entrainment-mixing broadening is only capable of producing drop diameters within this approximate range, or the methodology fails to adequately capture a signal from holograms containing $D_{max}$ exceeding 45 µm due to the relatively small sample size of such holograms (92). Using all available flight legs only increases the available number of holograms by 40. The relation of these trends to the presence of drizzle will be revisited later on.

Findings so far have displayed results for all flight legs in combination. However, the isolated large drop trend may be a consequence of macro-scale features associated with given flight legs (e.g., aerosol fields), and trends in Fig. 7 may simply correspond to the respective flight legs while other trends are epiphenomenal. To explore this possibility, the remaining results will look at how the isolated large drop trend is associated with different flight legs. Figure 8 shows HILD – OTHER differences similar to Fig. 7 but for separate flight legs (rows). The columns show HILD – OTHER for different variables. As in Fig. 7, samples are restricted to RF02A,B and RF10A,B due to the notably greater number of available holograms from these flight legs. This is because it is difficult to determine which flight legs have the most "notable" isolated large drop trend for flight legs with varying numbers of in-cloud samples, since more holograms will be pulled from flight legs with more measurements. Histograms of the variables' occurrence frequencies in Fig. 8 are shown for each flight leg separately in Supplementary Fig. S4.





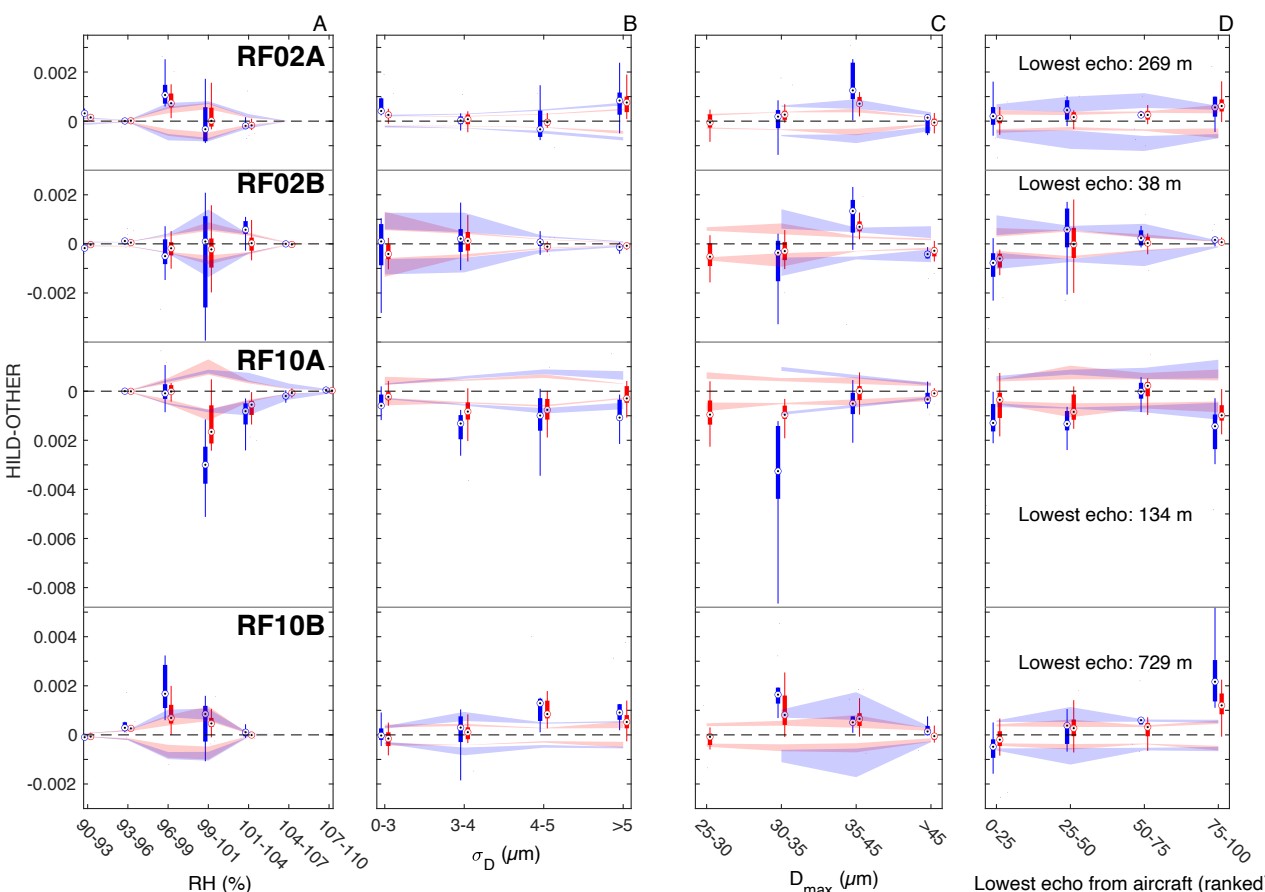

**Figure 8: A) Similar to Fig. 7 except shown for separate flight legs (rows). The panels show HILD – OTHER differences for *RH* (A), $\sigma_D$ (B), $D_{max}$ (C), and ranked distance from the aircraft to the lowest return echo (described in the text; D). Distances in D) are ranked from 0 (highest altitude) to 100 (lowest altitude) for each respective flight.**

Focusing first on results for *RH* (Fig. 8A), positive HILD – OTHER are only observed in the bin of 96–99% for RF02A and RF10B, the two flight legs associated with the isolated large drop trend (Fig. 6). Results shown for $\sigma_D$ (Fig. 8B) similarly show positive HILD – OTHER for the largest $\sigma_D$ values for these two flight legs. This is similarly restricted to the two flight legs associated with the isolated large drop trend. This suggests that the isolated large drop trend is related to the reported microphysical properties and not macro-scale features of the flight legs. Results for $D_{max}$ (Fig. 8C) also capture positive HILD – OTHER for $D_{max}$ from 35–45 µm for these two flight legs, although it is also captured for RF02B (2nd row). The trend is relatively weak in RF10B, especially since positive HILD – OTHER are also seen for $D_{max}$ from 30–35 µm (4th row). This weak trend may be due to the fact that drizzle is weakening the signal, since a clear precipitation signal of this flight leg visible from radar imagery (not shown). This is possibly suggested by the differences in boxplots for the $D_{max}$ datasets >25 µm and >30 µm. The positive HILD – OTHER signal is weaker when selecting among the $D_{max}$ >30 µm dataset, which



could be selecting from a greater number of samples influenced by descending drizzle drops/collision-coalescence compared to the $D_{max}$ >25 µm dataset.

To test this assumption, results in Fig. 8D are shown for the lowest radar detection of cloud, and the values are ranked from 0 (highest altitude) to 100 (lowest altitude) for each respective flight. We assume regions of the cloud with the lowest reaching condensate are likely associated with those producing the largest drops, since they will take longer to evaporate during descent. We also assume this variable has a greater "memory" for precipitation-initiation detection, since relatively larger drops are often either swept in/out or fall in/out of any given hologram. The lowest altitudes detecting condensate are shown in meters for each respective flight, with the two lowest values from the flight legs associated with the isolated large drop trend (top and bottom rows). Values are ranked since ranges of the lowest condensate vary considerably among the flight legs (Fig. S4D), and ranking also produces uniform distributions so that 25% of samples are within each of the four bins. Note that positive HILD – OTHER are observed at the largest ranked values for the two flight legs associated with the isolated large drop trend (8D; top and bottom row). The largest difference is observed for RF10B which far exceeds the uncertainty bounds, the flight leg with condensate reaching the lowest altitudes as well as the clearest precipitation signal amongst the four flight legs from visible inspection of radar reflectivity (not shown).

## 5 Implications of findings

The cumulative results from Fig. 7&8 suggest that the isolated large drop trend is related to entrainment-mixing, and presents evidence for entrainment-mixing drop size distribution broadening and its relevance for precipitation initiation. Alternative precipitation initiation mechanisms consistent with the paradigm of isolated large drops include "Ostwalt ripening" (e.g., Çelik and Marwitz 1999; Wood et al. 2002) and the turbulent sorting of droplets enhancing supersaturation within regions of high vorticity (Shaw et al., 1998). However, neither mechanism accounts for the isolated large drop trend occurring primarily in subsaturated conditions. Thus, results appear to be most consistent with entrainment-mixing drop size broadening.

At first consideration, the isolated large drop trend could solely be the result of smaller drops preferentially evaporating due to their greater surface area to volume ratio relative to larger drops (where drops are impacted by micro-scale temperature/vapor fields). However, holograms experiencing this trend are also associated with broader drop size distributions (Fig. 7E) and larger drops than holograms not exhibiting this trend (Fig. 7F). Further, this trend is associated with portions of the cloud where precipitation/condensate reaches the lowest altitudes from the respective cloud (which may invalidate the presence of Ostwald ripening assuming the parcels' locations vary considerably over timescales on the order of hours; Fig. 8D). The totality of findings here is therefore consistent with entrainment-mixing drop size broadening as a relevant precipitation-initiation mechanism (e.g., Lasher-Trapp et al. 2005; Hoffmann et al. 2019). It should be acknowledged that the bottleneck size range is loosely defined among previous studies, often defining the lower bound diameter ranging between 20–40 µm and the upper bound diameter ranging from 40–80 µm (Glienke et al., 2017; Grabowski and Wang, 2013; La et al., 2022; Pruppacher and Klett, 1996; Taraniuk et al., 2008). Results here are restricted primarily to



the lower bound, due to the low sample size of drops at the larger end of the bottleneck range. Because of this, results may primarily capture a diffusional-growth process only relevant to relatively smaller bottleneck droplets — and caution should be taken before disregarding alternative precipitation-initiation mechanisms not associated with the isolated large drop trend. A sensitivity test slightly increasing the guardrail dimensions to be 0.4 cm × 0.4 cm × 9.8 cm (and accordingly the outer sample volume increased to 0.7 cm × 0.7 ×10.1 cm) was performed with the hope of sampling more/larger bottleneck drops.

Unfortunately, due to the commonly observed quasi-exponential decrease in drop concentrations with increasing drop size beyond $D{\sim}20$ µm (shown for drops having high and low DCFs in Fig. 4A1), an "adequate" number of large bottleneck drops was still not captured. Additional sensitivity tests/suspected uncertainties are discussed in Appendix B.

## 6 Concluding remarks

  The purpose of this study is to present a new analytic paradigm for exploring possible correlations between drop clustering
and drop size, which are theorized to be associated with varying microphysical processes such as precipitation initiation. A novel method is introduced here to evaluate drop clustering on a drop-by-drop basis, in contrast to previous studies evaluating drop clustering in relation to its "system" (i.e., sample volume). Measurements are acquired using the HOLODEC probe, which is ideal for evaluating drop clustering due to the constant 3D sample volume size for each hologram. All droplets are first labeled as either (1) having a significantly high number of neighboring drops, (2) being significantly
isolated from other drops or (3) meeting neither condition. Then Monte Carlo simulations are run, randomizing the drop sizes in each respective hologram while preserving the drop locations. This allows us to determine likelihoods of drops having a given size as being either associated with a significant number of neighboring drops or as being significantly isolated from other drops. Results here show drops having sizes within the bottleneck size range ($D{\sim}25$–50 µm) are most likely to be significantly isolated from the nearest drops.

Additional analysis is performed to determine under which conditions holograms are associated with this "isolated large drop" trend. Holograms in subsaturated conditions and having low drop concentrations are associated with this trend, both of which suggest the isolated large drop trend is directly related to the presence of entrainment. These standalone findings could simply be the result of smaller drops preferentially evaporating due to their greater surface area to volume ratio relative to larger drops. However, holograms associated with the isolated large drop trend also have broader drop size distributions,
larger maximum drop sizes and also tend to overlie regions where cloud condensate/precipitation reaches the lowest altitudes. The totality of these findings suggests entrainment-mixing drop size broadening is a relevant precipitation initiation process, which has been suggested in previous modeling studies highlighting the ability of entrainment-mixing to produce "precipitation embryos" (e.g., Hoffmann et al., 2019; Lasher-Trapp et al., 2005). Notably, the isolated large drop trend is localized to a few flight legs, which may suggest its relevance is restricted to condition(s) beyond those analyzed here (e.g.,
background aerosol properties).

  Previous studies using similar holography measurements have found no correlation between the occurrence of drops in the bottleneck size range and drop clustering when diagnosed over individual sample volumes (e.g., Glienke et al., 2020; La et





al., 2022; Thiede et al., 2025). However, this study differs from past studies by not diagnosing the clustering of a collection of droplets, but rather determining the likelihoods that drops are either significantly isolated or surrounded by a significant number of droplets. This methodology therefore allows for the ability to evaluate "sub-volume" drop spatial inhomogeneities and can potentially capture signals of interest in Poisson distributed environments. An additional notable departure from these past studies is that the particle-by-particle clustering is only diagnosed on spatial scales of millimeters, whereas those studies also consider centimeter scale drop spatial inhomogeneities when diagnosing clustering.

Although CSET provides the largest publicly available dataset of processed HOLODEC measurements currently available (acknowledging 3D drop spatial coordinate data of HOLODEC measurements from the Aerosol and Cloud Experiments in the Eastern North Atlantic (ACE-ENA) campaign are unavailable), findings are still limited by the relatively low number of holograms available from the campaign. Uncertainties are further highlighted by the relatively few drops within the bottleneck size range, whose concentrations commonly decrease quasi-exponentially with increasing drop size beyond ~20 μm for standard in-cloud measurements. Applying the proposed methodology to a greater number of holograms from a greater number of environments is crucial towards continued evaluation of drop clustering in relation to drops in the bottleneck size range. Additional future work should be put towards constraining the entrainment-mixing broadening mechanism and its relevance pertaining to precipitation initiation.

**Appendix A: Illustration of Monte Carlo-Hologram Comparison methodology**

The Monte Carlo-Hologram Comparison methodology discussed here will follow step-by-step those listed in Table 3 (Section 3.2). Figure A1 shows multiple panels which aid in illustrating the methodology.



**Figure A1: A) Number of simulations in the HILD (solid lines) and OTHER (dashed lines) categories for all batch sizes. Results are shown for the $D_{max}$>25 µm (red lines) and $D_{max}$>30 µm (blue lines) datasets. Shading corresponds to batch sizes which are included in the analysis in Section 4.2. B) Normalized occurrence frequencies of *RH* samples shown from all HILD (red line) and OTHER (blue dashed line) simulations for the 800 hologram batch size from the $D_{max}$>30 µm dataset. C) the difference of the HILD and OTHER normalized occurrence frequency over all bins. Results are shown for all batch sizes from the $D_{max}$>30 µm dataset, where increasing warmer lines correspond to larger batch sizes. D) HILD – OTHER for the 96%–99% bin (purple shading in (C)) for all batch sizes included in the analysis. Results are shown for the $D_{max}$>25 µm (red line) and $D_{max}$>30 µm (blue line) datasets. The shading shows a range of confidence intervals, where the uppermost portion corresponds to the 95th percentile degree of significance**





**and the lowermost portion corresponds to the 80th percentile. Uncertainty is determined using a permutation method, where samples are randomly sampled from the HILD and OTHER datasets 10,000 times.**

This methodology incorporates a Monte Carlo sampling method in order to determine which holograms are associated with the isolated large drop trend. A number of holograms (i.e., batch size) are sampled randomly 20,000 times from the available dataset. This is done for two datasets: 1) all holograms with $D_{max}$>25 µm and 2) all holograms with $D_{max}$>30 µm, to check for consistency in the reported trends. Since there is no prior information available for what batch size is most appropriate to capture the isolated large drop trend, the batch size varies from 100 to 2200 holograms (Step 2 of Table 3). All of the simulations are classified as either possessing the isolated large drop trend (HILD), or as a category indicating the trend is not captured with high confidence (OTHER). Conditions for these classifications are shown in Step 3 of Table 3, where percentiles are determined following the Monte Carlo-DCF methodology. Note also that these conditions correspond to the same bottleneck drop size ranges as the large drop size bins from Fig. 4–6.

Figure A1A shows the number of simulations meeting the HILD and OTHER classifications (solid and dashed lines, respectively) for the $D_{max}$>25 µm and $D_{max}$>30 µm datasets (red and blue lines, respectively). Results show the number of HILD (OTHER) simulations increases with increasing (decreasing) batch size. Note that the smallest and largest batch sizes will have relatively few simulations meeting one of the category types. These simulations are not considered in the methodology, and only batch sizes having a minimum of 100 simulations meeting both categories are used in the analysis (Step 4 of Table 3). This is shown by the red and blue shading for the respective $D_{max}$ datasets. These simulations are not considered in order to increase the empirical power of the simulation sets (i.e., number of simulations rejecting the null hypothesis divided by the total number of simulations). Note the arrow in Fig. A1A points to the batch size with the maximum empirical power for the $D_{max}$>30 µm dataset (800), which will be referenced below.

For each batch size, the holograms from all the simulations meeting either HILD or OTHER conditions are combined to produce normalized frequency distributions for their respective categories (Step 5 in Table 3). To illustrate this, distributions for *RH* are shown for HILD and OTHER simulations in Fig. A1B (red and dashed blue lines, respectively) for the 800 hologram batch size and $D_{max}$>30 µm dataset. Due to the hypothesized impact of *RH* on the isolated large drop trend (and validated in Section 4.1), we focus on *RH* for an exploratory analysis of the Monte Carlo-Hologram Comparison methodology. The distributions are nearly identical due to the randomized hologram selection method, as well as the inability to discern which holograms contribute to the isolated large drop trend and which holograms do not. However, differences become apparent when taking the difference of HILD and OTHER distributions over each respective bin (Step 6 in Table 3), which is shown in Fig. A1C. Results here are shown for all simulation sets from the $D_{max}$>30 µm dataset with batch sizes meeting the empirical power condition in Step 4 of Table 3 (i.e., blue shading in Fig. A1A), where simulation sets with smaller (larger) batch sizes have colder (warmer) lines. Amongst all the simulation sets, there is a peak in the HILD – OTHER difference at *RH* of 96–99% (purple shading). This is consistent with results in Fig. 5, where the isolated large drop trend is observed for subsaturated conditions. Figure A1D shows HILD – OTHER of the 96–99% *RH* bin for all batch



sizes, corresponding to the values in the purple shading of Fig. A1C. Unlike Fig. A1C, results from the $D_{max}$>25 μm bin are also included (red line). Uncertainty bounds are also shown, using a permutation method where 10,000 simulations randomly sample the same number of holograms in the HILD and OTHER simulation sets and then similarly computing the difference of their distributions to discern statistical significance. The shading shows confidence intervals where the upper boundary corresponds to the 95th percentiles and the lower boundary corresponds to the 80th percentiles. Note that only 50% (75%) of

differences for the $D_{max}$>30 μm ($D_{max}$>25 μm) batch sizes have differences exceeding the 95th percentile, seen by overlying stars above the respective batch sizes. Once again, this is likely due to the random selection of holograms and the inability to discern which samples are inherently associated with the isolated large drop trend. For this reason, the lower boundary of 80th percentiles are used in Section 4.2 to test for statistical significance.

**Appendix B: Uncertainties and Sensitivity Tests**

Due to multiple components involved in the methodology, numerous sensitivity tests were devised to test the robustness of the isolated large drop trend. Supplementary Table S1 lists all such tests, and describes the rationale of the respective tests. Select uncertainties are further discussed here and are separately discussed for (1) uncertainties related to instrumentation and methodology and (2) uncertainties inherent in atmospheric phenomenon.

First considering uncertainty related to instrumentation uncertainty: the HOLODEC is associated with detection uncertainties

of the spatial coordinates of drops: which are 0.001 cm in the x- and y-dimensions, and is maximized in the z-dimension (the longest dimension) at distances of 0.01 cm (Yang et al., 2005). However, 0.01 cm is precisely the interval range for incremental shell size increase, so uncertainties are not expected to significantly impact results.

The shell sizes are determined based on the HOLODEC sample volume size and consideration of drop concentrations in the sample environments. Specifically, maximum shell sizes are determined by considering the hologram sample volume size

and the minimum shell sizes are determined by an intuitive consideration (i.e., crude consideration of counting statistics) that greater drop concentrations will allow increased resolution of DCF at smaller distances from the drops. Testing for different maximum shell sizes is particularly relevant since major trends discussed here occur at lower drop concentrations and for drops with the greatest isolation from neighboring drops (i.e., for the largest shell sizes). Unfortunately, increasing the shell size requires shrinking the guardrails which decreases the total number of available drops. However, sensitivity tests where

the maximum shell size is increased to 0.16 cm and 0.17 cm still capture the isolated large drop trend (not shown).

Shifting towards potential uncertainties related to cloud processes: similarly sized drops should follow similar trajectories, particularly drops within the stokes regime (D<~60 μm) where drop trajectories are partially a function of their size (Rogers and Yau, 1996). We can speculate that drops within this regime will be more likely to follow similar trajectories as similarly sized drops, and ultimately neighbor each other. This would result in smaller drops more likely to have neighboring drops,

since these drops are more numerous than drops in the bottleneck range. Results evaluating the likelihood of drops having different sizes neighboring the inner drops (i.e., the likelihood of drop sizes within the shells of drops having a specified drop size range) is shown in Supplementary Fig. S5. Evidence can be seen for drops neighboring those of similar sizes,





particularly for drops having D~20 µm (Fig. S5A). However, we suspect this is not a source of error/bias since if this were the case, the isolated large drop trend would also be observed (if not more likely observed) for holograms having high drop

concentrations as well as those not limited to subsaturated environments. Further, the trend is observed at low drop concentrations where a greater number of similarly sized large drops are observed.

We suspect the largest uncertainty arises from the limited dataset of available holograms. Holograms available for the analysis are taken from twelve flight legs within nine research flights, and 75% of the holograms are from four flight legs within two research flights. Another factor not explored here is the relation of these bottleneck trends to aerosol-cloud

interactions. While broad aerosol characteristics can be determined for given flight legs, the inability to diagnose such characteristics in-cloud due to droplet contamination is a major limiting factor.

**Code and Data Availability Statement**

All datasets from CSET are available to the public at data.eol.ucar.edu. MATLAB code for producing simulations and performing statistical tests is available by request.

**Acknowledgements**

We thank all those who gathered, worked with, and provided data from the CSET field campaign. We acknowledge support from the U. S. Department of Energy Atmospheric System Research (DOE ASR) through grants DE-SC0020134 and DE-SC0021103.

**Competing Interests**

The authors declare that they have no conflict of interest.

**Author Contributions**

J.D. designed the research methodology and performed the data analysis, with significant contributions from the coauthors.

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
