# Peer review of "Drop clustering and drop size correlations from holographic imagery suggest cloud droplet spectral broadening via entrainment-mixing"

_EGUsphere, 2025_

## Referee Comment (RC1)

**Referee Report**

**General assessment**

This paper analyzes a valuable dataset of airborne holographic measurements by applying a droplet cluster field (dCF) method to study small-scale clustering. The approach is clearly explained and the topic is important for understanding cloud microphysics. The authors results based on their analyses are that drops having sizes within the bottleneck size range (D of 25-50  $\mu$ m) are most likely to be significantly isolated from the nearest drops. Additional analyses were performed to determine under which conditions holograms are associated with this "isolated large drop" trend. They found that holograms in subsaturated conditions and having low drop concentrations are associated with this trend.

My understanding is that the main goal of the study is to detect *localized clustering* that may not be detectable using traditional system-wide statistics such as the spatial pair correlation function (or radial distribution function, g(r)). In a finite sample from a Poisson process (random droplet locations), one often observes patches that appear denser and others that appear sparser, simply due to statistical fluctuations. Locally, such fluctuations can look like clustering even though, globally, g(r) averages to 1.

The authors therefore aim to identify clustering that varies from place to place within the sample, possibly depending on droplet size. For example, size-dependent clustering would not be captured by a global g(r) unless the analysis were stratified by droplet size. My understanding is that the authors' method is essentially a localized version of g(r): a droplet-centric or neighborhood-based clustering measure, rather than a single system-wide statistic.

However, it is not entirely clear why the authors did not simply compute system-wide g(r) for subsets of droplets grouped by size. Such a modification of the traditional RDF approach could also reveal size-dependent clustering. Clarifying how the dCF method differs from (and improves upon) this more conventional strategy would help the reader understand the rationale for introducing a new diagnostic.

Finally, to this reader, it seems challenging for the authors' local metric to provide a robust signal based on only a few neighboring droplets around each primary droplet. A brief discussion of how statistical noise is controlled or mitigated in such local estimates would strengthen the paper.

**Major comments**

1. For a non-statistician such as this reviewer, sections 3 and 4 of this manuscript are mostly incomprehensible. Table 2, although presumably intended to clearly describe the classification of droplets into high or low DCFs, makes this reviewer uneasy due to what appears to be ad hoc additional classifications for high and low DCFs. This

reviewer would appreciate a demonstration of the method applied to an artificial dataset for which the clustering characteristics are specified. My concern is that achieving statistical significance based on local clustering that involves a very small number of particles could be difficult. Is the dataset large enough to overcome this difficulty? How can you demonstrate that?

**Minor comments**

- 1. lines 9-15. This could be misleading because it sounds like clustering on mm scales leads to enhanced collision rates and therefore the bottleneck solved. Therefore, you introduce a particle-by-particle neighborhood-counting method to evaluate these clustering trends. But this is not actually the same thing because the RDF at contact is the spatial factor relevant to collision rates. The neighborhood-counting method gives a different kind of clustering information (local heterogeneity), not the RDF itself. This could be revised as follows (for example): The question of how droplets rapidly grow large enough to initiate collision-coalescence has persisted for decades. Theories suggest that enhanced droplet clustering on millimeter scales can increase collision rates for droplets in the 'bottleneck' size range (~ 25-50 ?m). To investigate spatial organization in more detail, we introduce a novel droplet-centric diagnostic that evaluates local neighbor counts and proximities (i.e., clustering fields) on a particle-by-particle basis. Although not equivalent to the RDF, this approach provides complementary information about the heterogeneity of droplet environments.
- 2. lines 73-77: It's worth noting the confidence range for RH, since it's very sensitive and critical for distinguishing between saturated and unsaturated air. My own reading and for Rosemount probes after careful calibration is  $\sim 0.2-0.3$  K. At best these uncertainties give  $\pm 3-4$  % uncertainty.
- 3. section 2.2.2: The choice of 7 shells should be defended—why 7? Is it based on prior work, a sensitivity check, or computational limits?
- 4. section 2.2.2: Clarify whether edge corrections are applied for each shell separately or in a single step.
- 5. section 3.1: I believe Section 3.1 is analogous to analyzing spatial droplet clustering, since you're removing the dependency on droplet size. This may be worth mentioning.
- 6. Make it explicit in figure captions whether counts are cumulative or differential.
- 7. line 179: Rephrase "the maximum distance at which two drops can neighbor" since this is in fact a minimum spacing.
- 8. Improve figure captions so that readers can directly link each plot to the formal dCF definition.

- 9. lines 198-200: A sentence is duplicated here.
- 10. Figure 3B: It is unclear what is plotted. The caption states that the y-axis is "The number of high DCFs, low DCFs and DCFs meeting neither category." If this is the case, then the y-axis must be a number per some range of N, but this range or bin size is not stated. However, the units shown for the y-axis, as well as the text, suggest it is a number concentration. Please clarify. Furthermore, the category 'Drops used in analysis' should be explained because it does not make sense.
- 11. Figure 3B: Each of the upsloping patterns of 'High dCF' red points begins at one of the values for minimum  $\psi$  for high DCFs: 110, 175, 230, and 280. As a result, the patterns are dependent on the definitions listed in Table 2. What is the basis of these categories?
- 12. lines 285-6: The expression "meaning the DCFs will be identical for each hologram" should be clarified. I think it means that DCFs will be unchanged within a hologram, and also by extension within the entire set of holograms, because DCFs depend only on droplet spatial coordinates.
- 13. lines 287-8: "DCFs of the actual drop sizes" is confusing because DCFs have nothing to do with drop sizes. Please clarify.
- 14. line 294: Not all of us are statisticians, so please define Type I error.
- 15. lines 299-300: "DCF-drop size relationships are then determined for this set of holograms using the Monte Carlo-DCF methodology from Section 3.1." I don't see the need for the Monte Carlo aspect in order to determine DCF-drop size relationships.
- 16. line 380: "Figure 5A,C shows results separated into relatively large regions (~100 m) of subsaturated and supersaturated conditions." Please remind the reader (or explain if it was not explained already) how you identified these large regions from holograms that are only a few cm in extent. Even if you used sequences of holograms, their properties cannot be assumed to be continuous between them.
- 17. Figure 4: I am sorry but this figure is incomprehensible to me. The text that describes it does not help me (lines 333-343). A problem I have is evaluating the role of randomness/noise in these plots. How do we know if any of what is shown has statistical significance?
- 18. Figures 4, 5, 6: I suggest connecting the dots in the percentile plots to make the patterns easier to see. For me, it helped a lot to do that.
- 19. lines 524-7: Clarify: Do you mean "preferentially completely evaporating" not "preferentially evaporating"? It is difficult to understand your next sentence otherwise: "However, holograms experiencing this trend are also associated with broader drop size distributions (Fig. 7E) and larger drops than holograms not exhibiting this

- trend (Fig. 7F)" which is the result to be expected if the most of the smaller droplets are NOT completely evaporated. Tolle and Krueger (2014, Figure 17) found that broadening by partial evaporation is common.
- 20. lines 524-7: It seems to this reviewer that coalescence growth could also explain the "isolated large drop trend." Especially since the "this trend is associated with portions of the cloud where precipitation/condensate reaches the lowest altitudes from the respective cloud." This may be worth mentioning if you agree.

---

## Referee Comment (RC2)

Review of: "Droplet clustering and drop size correlations from holographic imagery suggest cloud droplet spectral broadening via entrainment-mixing"

Submitted for Consideration of Publication in Atmospheric Chemistry and Physics

John J. D'Allesandro, Robert Wood, and Peter N. Blossey

November 2nd, 2025

**Summary and Overall Evaluation**

This manuscript extends the existing literature that investigates sub-cm scale droplet clustering using data acquired by holographic imaging devices. The authors' primary conclusion is that within a few cloud trensects during the CSET campaign, HOLODEC data reveals a statistical tendency for large (> 25  $\mu$ m) cloud drops to be spatially isolated. In addition, this tendency seems to be related to other bulk variables (e.g. low local number concentrations and sub-saturated conditions) which the authors argue is more consistent with entrainment-mixing than other bottleneck growth hypotheses.

In general, I am pleased to see the HOLODEC data from the CSET campaign further investigated; this is a very rich important dataset that – at least in this reviewer's opinion – has not been sufficiently discussed in the literature to date. The advantages of using instruments that detect 3-dimensional particle positions are clear and papers that extend existing techniques to more effectively leverage this information are important and needed.

However, in this reviewer's opinion, there are significant improvements that are needed to make this work publishable in ACP.

**Comment on Previous Review**

Although I read this paper several times prior to writing this review, immediately preceding writing this document I also glanced at the public review supplied by Anonymous Referee #1 (posted on October 8th). Since the task of responding to critical reviewer comments often involves trying to accommodate differing (and sometimes orthogonal) opinions, let me state from the outset that I am in agreement with all elements of the general assessment and major comments (and most of the minor comments) provided by this first Anonymous Referee. I believe addressing the concerns raised by that referee are necessary for publication of this manuscript and my additional comments below are meant to augment and extend those criticisms.

**Additional Major Comments**

The authors make the claim that the approach introduced here is novel and, though this particular implementation is new, the approach builds on previously published approaches that are not yet cited. (For example, the idea of building out statistics based on separation distance from an existing droplet has striking

similarities to the conceptual starting point of the drop-size-dependent droplet clustering work using fractal and multi-fractal paradigms (see, e.g., [3, 8]). Further, the idea of using the local environmental conditions to characterize conditions under which the bottleneck problem may be better addressed was looked at in a way with strong connections to this approach in [13]. Finally, the idea of looking at averaged-local environments and combining a variety of scales rather than a scale-localized measure like g(r) has been explored repeatedly including through the use of the so-called volume-averaged pair-correlation function, Fishing statistic, Clustering index, Scaled Clustering Index, and through other common second-order statistics through the correlation-fluctuation theorem (see, e.g., [1, 2, 4, 5, 7, 11, 12]).

There are already several confusions associated with these tools and how they appropriately combine scaling information prevalent, but prevailing analysis suggests that aggregate statistics corresponding to a bulk statistic measured on scale R results from an unequally-weighted contribution from all scales less than or equal to R; the nature of this scaling, however, depends on the number of dimensions related to the observation, with different weighting factors existing in 1d, 2d, and 3d systems (see, e.g., [6]). Because of this, I am glad to see that the authors do not attempt to claim their approach identifies the key spatial scales of droplet clustering, but rather just detects whether individual drops are statistically isolated.

It is true that the majority of the above materials refer to analysis of optical-array-probe or similar 1-dimensional data, and thus the previous approaches have to be adapted to the three-dimensional domain – which is one of the contributions fo this manuscript. That being said, the approach outlined here very much is related to these previous works. To avoid confusion and to provide a bridge to previous work, I would encourage the authors to go back to those manuscripts to see if the language, notation, and approach can be adapted to align with already-existing frameworks.

Indeed, the authors' stated goal of "....determin[ing] the likelihood that drops of a given size will have a significantly high number of drops surrounding them as well as likelihoods that they are significantly isolated from neighboring drops" (lines 51-53) is *very* similar to the spirit of what the clustering index, Fishing statistic, volume-averaged pair-correlation function, etc. intend to do – determine if there is net clustering associated at or below some spatial scale. As the other reviewer suggested, the degree to which this approach is fundamentally different than a size-constrained version of previously-existing measures derived from and/or related to the radial distribution function is not clear. Although the authors of this manuscript suggest that the radial distribution function characterizes volumes, it calculates derived statistical quantities based on counting particle pairs separated by specified distances and thus every component of the sum in equation (1) ultimately is taken over an individual particle location. The stated different goal of doing this centered on a "per particle" basis vs. "per volume" basis claimed/implied by the authors may be a distinction without a true difference – especially as the metric utilized is ultimately found by adapting a single term of the sum utilized in calculating the RDE

- 2. This is, in spirit, a continuation of the above point but it also bears mentioning that the work presented here also has links to existing work on nearest-neighbor and/or Voronoi analysis methods especially in cases where the closer radial annuli have greater importance in the "local concentration/clustering" metric (as seems to be the case here if I am reading sections 2 and 3 correctly). This also links back to a different way of conceiving the radial distribution function as the sum of all of the *k*th nearest neighbor distributions (see, e.g, [9, 10]).
- 3. Despite rereading this paper multiple times (and being quite familiar with the general subject of droplet clustering statistics), all elements of figures 4, 5, and 6 are confusing to me. It seems like these figures provide a significant and necessary part of the central argument but I cannot determine whether I am interpreting these results correctly. Figures 7 and 8 are slightly easier for me to comprehend, though I am still a bit stymied

by how to interpret the uncertainty envelopes. The authors *do* attempt to describe all of these figures in detail, but I am unable to follow the explanations despite multiple attempts. I strongly suggest reconsidering whether there is an alternative way to visualize and/or explain these results that is more intuitive to a fresh reader.

- 4. To echo and extend a comment made by the first anonymous reviewer, the information and extensive set of logic and parameters presented in table 2 and table 3 are dizzying to a reader not yet fully comfortable with this analysis technique. The authors specifically talk about the desire to avoid analysis techniques that invite "p-hacking". The introduction of *so many* parameters associated with the technique raises the question as to whether or not the conclusions or quantitative results may be sensitive to the choice of these (what seem to be) arbitrary parameters associated with the method. For example why 7 different radii? Why these cutoffs for N for ψ and shell size (table 2), why these cutoffs for HILD determination (table 3), why the batch sizes used (and would this matter if the instrument cadence were different like new developing holographic systems)? What here might depend on the number density of observations? What would change for clouds that are not stratocumulus or for larger fields-of-view or for two-dimensional PIV images? The claim is explicitly *not* that the authors chose parameters explicitly to get a significant result, but rather the (arguably equally important) point that to apply this technique more broadly to other campaigns, instruments, systems, or environments the authors provide little guidance to the reader how to adapt their methodology more broadly. To be explicit, my concern here is twofold: (i) the methods, though explained in detail, are confusing to the
  - To be explicit, my concern here is twofold: (i) the methods, though explained in detail, are confusing to the reader. Hopefully the general approach could be explained in a simpler manner and the finer details could be moved to an appendix or supplement?, and (ii) as a reviewer who is tasked with validating the soundness of the approach as well as the veracity of the results, there are just too many unjustified and (as far as I can tell) unphysically motivated parameters associated with the method to be sure that the conclusions are not a spurious consequence of the specific parameters used/chosen.
- 5. One issue that may be of foundational concern are we certain that the HOLODEC retrieval accuracy for droplets near an existing large droplet is equally effective as it would be in that region of the detector if the large drop were not present? HOLODEC reconstructions are built upon an algorithm that very easily could have lower effectiveness for other droplet detections in the vicinity of a large drop, which could mean that the entirety of this analysis could be the result of a spurious instrumental anomaly. This concern isn't merely academic although I am not an expert on the details of in-line holographic reconstruction, it seems entirely reasonable that like in other optical reconstructions a small signal near a large signal could be missed or falsely attributed to a fluctuation related to the large signal. Before making claims about the fundamental microphysics and precipitation initiation mechanisms, it seems quite important to ensure that the instrument has a quantum efficiency of small drop detection that does not depend on the presence or absence of a large drop nearby.

Given that it seems the statistical evidence is subtle, it bears pointing out that even if the sensitivity of small particle detection being impaired by the presence of a large particle nearby only affected part of the field of view that was retained for use in this study, that could very easily induce the entire statistical signal studied here.

Unfortunately, I don't believe the Monte Carlo mechanisms in place throughout this paper are designed in such a way as to verify or reject this concern. At the *VERY* least, in line 358, the authors should modify the text to read "...drops within the bottleneck size range are most likely to be measured to be significantly isolated from surrounding drops". However, I do not think that merely adding this language is sufficient to ignore this overarching concern. A skeptical reader may be unconvinced by the entire message of the

paper unless evidence that small particle detections near large particles are not inhibited in the holographic reconstruction.

**Additional Minor Comments**

1. Although this is a minor comment, I believe it is an important one. Much of the work over the last 20+ years on cloud particle clustering has been obfuscated because of the lack of a consensus and author-to-author consistency regarding terminology. Tying what has been done to previous work is particularly important (which is the foundation underlying my first two major comments), but taking care to do so in a way that doesn't further confuse things is also important. As such, I implore the authors to choose a different notational convention than to use the idea of " $g_{drop}$ " (e.g. see equation (3)). Multiple papers already have been written about the conditions as to when g(r) can and cannot be calculated and still contain the physical meaning ascribed to it, and many of those conditions are not easily mappable to a particle-by-particle approach as presented here. (For example, although g(r) can always be calculated from any existing data-set, the underlying assumption of spatial homogeneity requires  $g(r \to \infty) \to 1$  and statistical viability requires the denominator to have sufficient expectation so that shot-noise doesn't dominate the signal. Some of these concerns may not persist for individual particle detections, but assuredly if the whole point is to determine whether a particle is statistically isolated the *premise* of the approach is that there is spatial heterogeneity on the scales of investigation. Although it may seem harmless to adopt the use of the statistical metric outside its formal range of validity, we have previously seen (see, e.g., [1] and the corresponding counter-examples presented in [7] that applying radial distribution functions outside of ther range of validity for homogeneous systems results in misunderstandings.)

The use of g(r) on a particle-by-particle metric invites further confusion, especially if the approach introduced here gains wider traction.

- 2. I would find it valuable if the total number of holograms were presented for each of the flight legs in table 1 as well, so the reader can get a sense of the accepted/utilized fraction of detected holograms in each of the flight legs.
- 3. The paper cited on line 109 by Thiede et al. does not use HOLODEC data. (The system used in that paper has some similarities to HOLODEC, but is mounted on an aerostat, takes observational data at a much higher cadence, and is processed slightly differently).
- 4. Given the fairly severe aspect ratio utilized (see line 119), there are questions as to the true three-dimensionality of the domain explored here. It is notable that this has a more extreme aspect ratio than other studies that have used the HOLODEC data, which is one of the reasons that the guard-area (or, as stated here, guardrail) approach has been largely abandoned in some other work. It is not clear how that may impact the final results.
- 5. Line 143 I don't believe the authors meant to say *i*th particles here.
- 6. Line 165 "counting statistic error" I'm not sure that "error" is the appropriate term in this context. "Uncertainty" or "bias" or "fluctuation" may be more appropriate here.
- 7. Line 179 "maximum" probably should be "minimum" here.

- 8. Lines 208-210 see major comment 2 related to similarities/links to Voronoi/nearest-neighbor techniques.
- 9. Line 222 In what way is 0.15 cm "ideal"? Would that parameter extend to other cloud conditions/instruments/etc? This links back to major comment 4.
- 10. Line 243 By choosing the same number of high and low DCFs to control for environmental conditions also seems like it introduces another scale parameter for the analysis, especially if this method were to be applied to a more heterogeneous domain in regards to number concentration or other parameters.
- 11. Line 246 "randomly" I'm not sure what is meant by this in this context.
- 12. Lines 470-473. The fact that holograms with  $D_{\rm max} > 45~\mu{\rm m}$  do not exhibit this trend is very much concerning and, to me, seems to run counter to the authors' explanation that this signal is likely induced by entrainment-mixing.
- 13. Lines 525-527. The fact that holograms with larger drops exhibit broader size distributions and fluctuations of other microphysical variables is more or less tautologically true, isn't it? (Especially if the instrument can only see droplets above a certain size reliably.)
- 14. Line 550 at the end of the sentence including numbers, it seems to be important to specify that these are relative to the drop's *local environment* and not in absolute/cloud-averaged terms. This local environment is tied to a specific measurement scale, and that is another spatial scale of relevance when exploring this question.
- 15. Line 554 "significantly isolated"; this raises the question as to whether being statistically significant is sufficient to affect localized dynamics/microphysical responses.
- 16. Line 570 I believe it is important at this stage to indicate the scale (sub-cm scale) where this conclusion is likely valid.

**References**

- [1] B. Baker and R. Lawson. Analysis of tools used to quantify droplet clustering in clouds. *Journal of the Atmospheric Sciences*, 67:3355–3367, 2010.
- [2] B.A. Baker. Turbulent entrainment and mixing in clouds: A new observational approach. *Journal of the Atmospheric Sciences*, 49:387–404, 1992.
- [3] Y. Knyazikhin, A. Marshak, M.L. Larsen, W.J. Wiscombe, J.V. Martonchik, and R.B. Myneni. Small-scale drop size variability: Impact on estimation of cloud optical properties. *Journal of the Atmospheric Sciences*, 62:2555–2567, 2005.
- [4] A.B. Kostinski and A.R. Jameson. On the spatial distribution of cloud particles. *Journal of the Atmospheric Sciences*, 57:901–915, 2000.
- [5] L. Landau and E.M. Lifshitz. Statistical Physics. Butterworth Heinemann, Oxford, UK, 1980.
- [6] M.L. Larsen. *Studies of discrete fluctuations in atmospheric phenomena*. PhD thesis, Michigan Technological University, 2006.

- [7] M.L. Larsen. Scale localization of cloud particle clustering statistics. *Journal of the Atmospheric Sciences*, 69:3277–3289, 2012.
- [8] A. Marshak, Y. Knyazikhin, M.L. Larsen, and W.J. Wiscombe. Small-scale drop size variability: Empirical models for drop-size-dependent clustering in clouds. *Journal of the Atmospheric Sciences*, 62:551–558, 2005.
- [9] B. Picinbono and C. Bendjaballah. Characterization of nonclassical optical fields by photodetection statistics. *Physical Review A*, A71:013812, 2005.
- [10] B. Picinbono and C. Bendjaballah. Poisson processes with integrable density. *IEEE Transactions on Information Theory*, 52(12):5606–5613, 2006.
- [11] R.A. Shaw. Particle-turbulence interactions in atmospheric clouds. *Annual Review of Fluid Mechanics*, 35:183–227, 2003.
- [12] R.A. Shaw, A.B. Kostinski, and M.L. Larsen. Towards quantifying droplet clustering in clouds. *Quarterly Journal of the Royal Meteorological Society*, 128:1043–1057, 2002.
- [13] J.D. Small and P.Y. Chuang. New observations of precipitation initiation in warm cumulus clouds. *Journal of the Atmospheric Sciences*, 65:2972–2982, 2008.

---

## Author Comment (AC1)

**Response to the reviewers**

We thank the reviewers for their time invested in the manuscript, and for their detailed thoughts and considerations. We believe the incorporated revisions have certainly improved the overall quality of this paper.

Concerning specific reviewer comments: The reviewers' comments are in black text, our comments are in red text and edits made to the manuscript are in blue text. All references to table and figure numbers correspond with the new manuscript version.

A major point from both reviewers was concerning the comprehensibility of the manuscript. Significant effort has been put towards revising the manuscript to improve clarity and general readability.

**Reviewer #1:**

We thank the reviewer for their insightful input and time invested towards reviewer this paper. We believe the manuscript has been thoroughly improved upon through addressing their comments. We address the author's comments underlying their respective points.

General assessment

This paper analyzes a valuable dataset of airborne holographic measurements by applying a droplet cluster field (dCF) method to study small-scale clustering. The approach is clearly explained and the topic is important for understanding cloud microphysics. The authors results based on their analyses are that drops having sizes within the bottleneck size range (D of 25-50 μm) are most likely to be significantly isolated from the nearest drops. Additional analyses were performed to determine under which conditions holograms are associated with this "isolated large drop" trend. They found that holograms in subsaturated conditions and having low drop concentrations are associated with this trend.
My understanding is that the main goal of the study is to detect localized clustering that may not be detectable using traditional system-wide statistics such as the spatial pair correlation function (or radial distribution function, g(r)). In a finite sample from a Poisson process (random droplet locations), one often observes patches that appear denser and others that appear sparser, simply due to statistical fluctuations. Locally, such fluctuations can look like clustering even though, globally, g(r) averages to 1.
The authors therefore aim to identify clustering that varies from place to place within the sample, possibly depending on droplet size. For example, size-dependent clustering would not be captured by a global g(r) unless the analysis were stratified by droplet size.
My understanding is that the authors' method is essentially a localized version of g(r): a

droplet-centric or neighborhood-based clustering measure, rather than a single system-wide statistic.

However, it is not entirely clear why the authors did not simply compute system-wide g(r) for subsets of droplets grouped by size. Such a modification of the traditional RDF approach could also reveal size-dependent clustering. Clarifying how the dCF method differs from (and improves upon) this more conventional strategy would help the reader understand the rationale for introducing a new diagnostic.

We thank the reviewer for their comments and agree with the assessment as a whole. Concerning the author's suggestion of computing system-wide g(r) for drops of different sizes: while it may be possible to do so with additional considerations, using the simple approach as suggested is associated with major counting uncertainties, which we demonstrate below. We compute g(r) as suggested by the reviewer for different drop size ranges. To do this, we must alter the RDF equation to compute a partial RDF, such that

$$g_{Drange}(r) = \sum_{i=1}^{N_{Drange}} \frac{\psi_i(r)/N_{Drange}}{(N-1)\left(\frac{dV_r}{V}\right)}$$

where $\psi(r)$ is the number of particles surrounding the $i$th particle within the surrounding spherical shell volume between radii $r - \Delta r/2$ and $r + \Delta r/2$, $V$ is the measurement volume over the entire hologram, $N$ is the number of drops within the guardrails and $dV_r$ is the measurement volume enclosed within shells having radii $r - \Delta r/2$ and $r + \Delta r/2$. The only new variable is $N_{Drange}$, which is the concentration of drops within the given size range.

Figure R1 below shows $g_{Drange}$ for drops with diameters less than 25 μm (red lines) and greater than 25 μm (blue lines). Results are only shown for flight leg RF10B (Table 1 in manuscript). The thick dark red and dark blue lines are the averages of all individual holograms' $g_{Drange}$, whereas semi-transparent red and blue lines are partial RDFs of individual holograms. Results are restricted to holograms containing drops with diameters (D) greater than and less than 25 μm.

[Figure]

Figure R1: Partial RDFs computed over the second flight leg from RF10 (RF10B; Table 1) separately for drops with D<25 μm (red lines) and D>25 μm (blue lines). A) Partial RDFs are

computed where the radial distance bins (r bins) having *g(r)* = 0 are omitted when averages are computed (thick lines). B) Average partial RDFs are computed where averages include radial distance bins having *g(r)* = 0. C) Average partial RDFs similar to A and B, but are only computed for the same number of drops with *D*>25 μm and *D*<25 μm. Drops for the *D* range with the greater concentration are randomly selected to be the same number of drops for the *D* range with the lower concentration. Solid (dashed) lines correspond with average partial RDFs which include (omit) *g(r)* = 0 bins when computing averages.

Figure R1a shows partial RDFs computed for drops having *D*>25 μm (blue lines) and *D*<25 μm (red lines), and averages for the respective drop size ranges are denoted by the thick, darkly shaded lines. Partial RDFs with g(r) approaching one is indicative of Poisson (randomly) distributed droplets, and greater (lower) values indicate whether more (less) droplets are observed at the given r distance. In contrast to the manuscript finding large drops are likely isolated, Figure R1a indicates large drops are *more* likely to be in close proximity to surrounding drops compared with smaller drops (the red line is higher than the blue line). However, the characteristic inverse exponential shape of the RDFs results from ignoring r bins where g(r) = 0. Figure R1b only shows the average partial RDFs of the two drop size ranges similar to R1a, except r bins with g(r)=0 are included in the averaging. Note that these results have averages centered at g(r) = 1 at the larger r bins similar to R1a (r > ~0.75 cm), but now g(r) decreases below 1 at small r bins.

This this is due to counting statistical uncertainties, which while referenced in the previous manuscript version, has been updated for further clarification. Namely, when there are no pairs counted ($\Psi$=0), the expression may arguably be invalid since there may be no data to be weighted by the denominator. This can be conceptually understood if we were to compute an RDF for droplets within a very small sample volume (e.g., our HOLODEC measurements of a few cubic centimeters) compared with a very large volume (e.g., much greater than a few cubic centimeters), where the latter will have a larger sample size of droplets and a greater likelihood of detecting droplet pairs at very close proximities to each other (i.e., very small r bins). This is particularly important to consider when exploring clustering characteristics at smaller spatial scales explored here (mm) than those where such biases are less likely to occur (cm).

It is also for this reason that Figure R1a&b show small droplets are more likely to be isolated from other drops than larger drops (blue lines above red lines). This is because there is usually a greater number of drops with D < 25 μm than drops with D > 25 μm, and this consequentially produces a greater number of g(r) = 0 to be included in the averages for D > 25 μm. To highlight this, Figure R1c shows average partial RDFs of the two drop size ranges but it is only computed for the same number of available drops with D > 25 μm and D < 25 μm for each hologram. Because there are usually more small drops than larger drops, partial RDFs are generally computed for a reduced number of small drops equaling the number of large drops. Average partial RDFs are shown when omitting g(r) = 0 bins (dashed lines) and including g(r) = 0 bins (solid lines) in the calculation. We see for both sets of average RDFs that the large drops are now more isolated than smaller drops (red lines above the blue lines).

Now that we have potentially captured a similar trend as in our paper, it follows that we would want to determine the statistical significance of these partial RDFs. However, there is no universally accepted method to do so. Previous studies have simulated Poisson distributed environments of droplets (Larsen and Shaw, 2018) or used a simple ad hoc methodology (Larsen et al., 2018). In contrast, our study uses a permutation method to determine uncertainty.

To conclude: we showed 1) that large drops can be shown to be more isolated than small drops using partial RDFs and 2) how statistical counting uncertainties can introduce errors into our outputs. However, our proposed methodology avoids these counting uncertainties by determining the likelihoods drops of a given size range will be 1) isolated or 2) have a high number of neighboring drops by randomly shuffling the drop sizes within the respective holograms while keeping the droplet spatial coordinates unchanged (Section 3.1 in the manuscript). Our methodology also tests for statistical significance whereas there is some ambiguity of how to best achieve this solely using partial RDF computations. We have made edits throughout the manuscript to further clarify the methodology, and have elaborated how our methodology avoids some contingencies of the partial RDF computations.

We also now include this response to the reviewer in the supplementary material to highlight some of the counting uncertainties we reference in the paper.

Finally, to this reader, it seems challenging for the authors' local metric to provide a robust signal based on only a few neighboring droplets around each primary droplet. A brief discussion of how statistical noise is controlled or mitigated in such local estimates would strengthen the paper.

We acknowledge the relatively few large droplets available in the dataset, which is why efforts were made to include as many holograms as possible (e.g., Table 1, Figures 3&4, Supplementary figure S4). While acknowledging the semi-exponential decrease in drop concentration with increasing drop sizes with diameters from 25–50 µm, thousands of holograms are available with drops having diameters ranging from 30–50 µm, and there are over 1200 drops with diameters from 37.5–50 µm. In fact, it was partly our intent that focusing on statistics of individual drops is an improvement in the counting statistics relative to a hologram analysis (greater number of drops compared to holograms).

Assuming the reviewer is addressing the few drops with diameters exceeding 30 or 37.5 microns (e.g., panels A1,B1,C1,D1 in Figures 3,4,5), we remind the reviewer that these drop size histograms only show the number of those with high and low DCFs. A notably high number of drops not meeting either the high or low DCF categories (green datapoints in Figure 3) exist. These drops not meeting the high or low DCF categories *are used* when determine the statistical significance of whether drops of a different size will be associated with high DCFs or low DCFs.

However, we worry our measures of statistical significance were difficult to interpret, in that we test for statistical significance in all of the presented results (Figures 3-7). We have made efforts

to clarify the method for testing significance (discussed below), including editing Table 2 to simplify the displayed methodology.

Major comments

1. For a non-statistician such as this reviewer, sections 3 and 4 of this manuscript are mostly incomprehensible. Table 2, although presumably intended to clearly describe the classification of droplets into high or low DCFs, makes this reviewer uneasy due to what appears to be ad hoc additional classifications for high and low DCFs. This reviewer would appreciate a demonstration of the method applied to an artificial dataset for which the clustering characteristics are specified. My concern is that achieving statistical significance based on local clustering that involves a very small number of particles could be difficult. Is the dataset large enough to overcome this difficulty? How can you demonstrate that?

We thank the reviewer for the comment. To begin, we quickly state that many of the Table 2 classifications were indeed determined ad hoc. We hope Figure S3 (previously Figure 3 in the original manuscript) addresses the validity of choices made in Table 2, since it displays how in each hologram, approximately 1/3 of drops are classified as having high DCFs and ~1/3 of drops are classified as having low DCFs. We have made updates to the manuscript discussing this was the intention when developing the DCF classification method.

Moving to the author's request, we now provide the author with an analysis of an artificial dataset to show how it compares with the real data. Namely, we use drop data from the same holograms and simulate Poisson distributed drop spatial coordinates for the holograms used from the four primary research flight legs focused on in Figures 5–7 (RF02A,B and RF10A,B; Table 1). Below, we compare DCF results from Figure 5 (identical to Figure R2 below) with Figure R3, which uses the simulated datasets

[Figure]

Figure R2: Identical to Figure 5 from manuscript

Figure R3: Similar to Figure R2 but for simulated data

Our results in Section 4 focus on flight legs RF02A and RF10B since they capture the isolated large drop trend (blue stars and diamonds for 25 µm<D<37.5 µm in Figure R2A and R2D). When applying our methodology to the simulated dataset (Figure R3), these trends are no longer observed, and only spurious cases are observed where either high or low DCFs are significantly likely to be associated with a drop size range.

We can similarly concentrate our results on droplets with the highest and lowest DCFs similar to Figure 3, which shows DCF statistics for increasingly isolated and increasingly clustered drops from panels A to B to D. Similar to Figure 3, Figure R4 shows that the increasingly most isolated drops are most likely to be large drops (blue stars and diamonds moving from panels A to B to D). Although panels A and B have the same number of blue diamonds for large drops, note that the percentile values (blue dots) increase in panel B for drop size ranges of 25–30 µm and 37.5–50 µm.

When applying the same methodology to the simulated data (Figure R5), this trend is not observed and only spurious drop ranges are associated with significant likelihoods of DCF categories. We conclude by saying that a major advantage of our methodology is that we are not required to make any assumptions of how droplets are oriented spatially when performing our drop clustering–drop size analysis.

[Figure]

Figure R4: Similar to Figure 3 from manuscript but only uses data from RF02A,B and RF10A,B

Figure R5: Similar to Figure R4 but for simulated data

We address the reviewer's final concern here (sample size of large drops), although we refer to our response to the last paragraph of the general assessment for specific drop statistics and additional discussion. While we note thousands of available droplets are available, we also wish to state that all large drops are used when determining DCF likelihoods. This is because our methodology of randomly shuffling drop sizes amongst the drops is done using all large drops regardless of what DCF category they are in.

Minor comments

1. lines 9-15. This could be misleading because it sounds like clustering on mm scales leads to enhanced collision rates and therefore the bottleneck solved. Therefore, you introduce a particle-by-particle neighborhood-counting method to evaluate these clustering trends. But this is not actually the same thing because the RDF at contact is the spatial factor relevant to collision rates. The neighborhood-counting method gives a different kind of clustering information (local heterogeneity), not the RDF itself. This could be revised as follows (for example): The question of how droplets rapidly grow large enough to initiate collision-coalescence has persisted for decades. Theories suggest that enhanced droplet clustering on millimeter scales can increase collision rates for droplets in the 'bottleneck' size range ($\sim$ 25-50 um). To investigate

spatial organization in more detail, we introduce a novel droplet-centric diagnostic that evaluates local neighbor counts and proximities (i.e., clustering fields) on a particle-by-particle basis. Although not equivalent to the RDF, this approach provides complementary information about the heterogeneity of droplet environments.

We thank the reviewer for the comment and agree it could unintentionally be interpreted as a reference to collision-coalescence related theories. We have rephrased this:

Line 11-13: The question of how droplets rapidly grow large enough to initiate collision-coalescence has persisted for decades. Many theories explaining the production of sufficiently large drops (i.e., those in the "bottleneck" size range; ~25–50 μm diameters) involve drop clustering on millimeter scales.

2. lines 73-77: It's worth noting the confidence range for RH, since it's very sensitive and critical for distinguishing between saturated and unsaturated air. My own reading and for Rosemount probes after careful calibration is~ 0.2– 0.3 K. At best these uncertainties give ±3– 4 % uncertainty.

We agree with the reviewer's assessment. Although we were unable to find any references to the official uncertainty for measurements from the CSET campaign, we have included this text in the manuscript:

Line 81-82: The combined uncertainties from the Rosemount temperature probe and VCSEL at the temperature range of observations in this study are expected to result in an RH uncertainty of ~5%.

3. section 2.2.2: The choice of 7 shells should be defended–why 7? Is it based on prior work, a sensitivity check, or computational limits?

The choice of 7 is chosen in combination with conditions for classifying high and low DCFs (step 3 in Table 2) to approximately produce terciles of drops having high and low DCFs, while also providing adequate spatial resolution for discretizing drops to avoid producing similar degrees of high and low DCFs (for the purpose of sorting drops). We have simply updated the text to address this as:

Lines 201-202: The number of shell sizes in our analysis is chosen in order to produce a reasonable resolution for determining spatial correlations between droplets.

4. section 2.2.2: Clarify whether edge corrections are applied for each shell separately or in a single step.

5. section 3.1: I believe Section 3.1 is analogous to analyzing spatial droplet clustering, since you're removing the dependency on droplet size. This may be worth mentioning.

While there is spatial droplet clustering being analyzed drop-by-drop, we are only determining spatial droplet clustering *in relation to drop size*. We are not removing the dependency on drop size, but rather simulating how the DCFs would be distributed if we were to randomly shuffle the drop sizes within their respective holograms. We then compare the observed drop size and their associated DCFs with the randomized drop sizes and their associated DCFs. We have made updates to Section 3.1 to improve clarity.

6. Make it explicit in figure captions whether counts are cumulative or differential.

We are unsure what the reviewer is referring to here. Does the reviewer refer to histograms in Figures 3–6? These histograms are differential, and do not have a characteristic cumulative distribution (approaching larger values as the x-axis value increases). We have not addressed this point, and would ask the reviewer for clarification if possible.

7. line 179: Rephrase "the maximum distance at which two drops can neighbor" since this is in fact a minimum spacing.

We thank the reviewer for pointing this out. It has been updated.

8. Improve figure captions so that readers can directly link each plot to the formal dCF definition.

We now spell out drop clustering fields (DCFs) in all figure panels which reference it.

29. lines 198-200: A sentence is duplicated here.

We thank the reviewer for finding this. We have corrected it.

10. Figure 3B: It is unclear what is plotted. The caption states that the y-axis is "The number of high DCFs, low DCFs and DCFs meeting neither category." If this is the case, then the y-axis must be a number per some range of N, but this range or bin size is not stated. However, the units shown for the y-axis, as well as the text, suggest it is a number concentration. Please clarify. Furthermore, the category 'Drops used in analysis' should be explained because it does not make sense.

Since each drop has one drop clustering field (DCF), "the number of high DCFs, low DCFs and DCFs meeting neither category" is equivalent to "the number of droplets having high DCFs, low DCFs and DCFs meeting neither category." We have updated the text as:

Line 620-622: The number of droplets having high DCFs, low DCFs and DCFs meeting neither category in each hologram ($N_{DCF\_category}$) related to their respective hologram's drop concentrations ($N$).

11. Figure 3B: Each of the upsloping patterns of 'High dCF' red points begins at one of the values for minimum ψ for high DCFs: 110, 175, 230, and 280. As a result, the patterns are dependent on the definitions listed in Table 2. What is the basis of these categories?

These categories and associated thresholds were selected ad hoc to classify ~1/3 of droplets as having high DCFs and ~1/3 of droplets as having low DCFs in each hologram (stated at line 214-224 and in Appendix A). We note that a sensitivity test where we slightly alter these threshold values is provided in the supplementary material (Table S1; discussed in Appendix C) and all major trends are still observed. We have also emphasized this DCF categorization in the revised manuscript.

12. lines 285-6: The expression "meaning the DCFs will be identical for each hologram" should be clarified. I think it means that DCFs will be unchanged within a hologram, and also by extension within the entire set of holograms, because DCFs depend only on droplet spatial coordinates.

This is correct. We now also added another sentence underlying it for clarification:

Line 290-291: Since DCFs remain unchanged, the same number of drops meeting the high/low DCF classifications are selected from each hologram.

13. lines 287-8: "DCFs of the actual drop sizes" is confusing because DCFs have nothing to do with drop sizes. Please clarify.

We agree this wording could be confusing, since drops have their own DCFs. We have reworded multiple sentences in this section for clarity, and this phrasing has been removed.

14. line 294: Not all of us are statisticians, so please define Type I error.

We now define it in the text:

Line 298-299: However, this requires the manual selection of holograms when exploring DCF–drop size relationships and may result in Type-I errors (i.e., falsely rejecting the null hypothesis).

15. lines 299-300: "DCF-drop size relationships are then determined for this set of holograms using the Monte Carlo-DCF methodology from Section 3.1." I don't see the need for the Monte Carlo aspect in order to determine DCF-drop size relationships.

Section 3.1 discusses how Monte-Carlo simulations are used to determine DCF-drop size relationships (this refers to the shuffling of drop sizes within a hologram while keeping the drop locations unchanged). We have made overall edits to the manuscript to improve the clarity of the methodology.

16. line 380: "Figure 5A,C shows results separated into relatively large regions (~100 m) of subsaturated and supersaturated conditions." Please remind the reader (or explain if it was not explained already) how you identified these large regions from holograms that are only a few cm in extent. Even if you used sequences of holograms, their properties cannot be assumed to be continuous between them.

We did identify how HOLODEC measurements are collocated with measurements from other instrumentation in Section 2.1. We now remind the reader:

Line 385-387: Holograms are selected from collocated 1 Hz RH measurements having relatively coarse spatial resolutions (~100 m; previously described in Section 2.1).

17. Figure 4: I am sorry but this figure is incomprehensible to me. The text that describes it does not help me (lines 333-343). A problem I have is evaluating the role of randomness/noise in these plots. How do we know if any of what is shown has statistical significance?

We have rewritten these lines to improve the clarity (shown below), as well as other passages related to this analysis. We explicitly state how statistical significance is defined using the Monte Carlo-DCF methodology.

Line 336-346: We first focus on the top left panels: Figure 3A shows percentile results for drop sizes categorized as having high and low DCFs amongst all holograms used in this study. A few "statistically significant" drop size ranges can be found, i.e., DCFs either below or exceeding the 5th and 95th percentiles, respectively (represented by the diamonds and stars). This includes a statistically significant drop size range from 30–37.5 μm. To better illustrate the comparison between the actual and Monte Carlo distributions, the 30–37.5 μm drop size bin from Fig. A1 is magnified and displayed in the middle of the figure. The drop size distributions from the Monte Carlo simulations can now be seen as the thin red and blue lines (although giving a partial appearance of purple lines due to their overlap), corresponding to the high and low DCFs, respectively. The actual number of drops having low DCFs (thick blue line) in this bin exceeds the number of drops having low DCFs for most of the Monte Carlo simulations (thin blue lines), specifically exceeding over 95% of them. We define cases where the actual number of counts in a drop size bin is greater (less) than those for 95% (5%) of the simulations as being statistically significant. Therefore, drops having low DCFs are significantly likely to possess D ranging from 30–37.5 μm.

18. Figures 4, 5, 6: I suggest connecting the dots in the percentile plots to make the patterns easier to see. For me, it helped a lot to do that.

We appreciate the recommendation from the reviewer and have updated the figures accordingly.

19. lines 524-7: Clarify: Do you mean "preferentially completely evaporating" not "preferentially evaporating"? It is difficult to understand your next sentence otherwise: "However, holograms experiencing this trend are also associated with broader drop size distributions (Fig. 7E) and larger drops than holograms not exhibiting this trend (Fig. 7F)" which is the result to be expected if the most of the smaller droplets are NOT completely evaporated. Tolle and Krueger (2014, Figure 17) found that broadening by partial evaporation is common.

We thank the reviewer for the comment and have updated the text to state it is preferentially completely evaporating:

Line 529-531: At first consideration, the isolated large drop trend could solely result from the removal of small drops via preferential evaporation due to their greater surface area to volume ratio relative to larger drops (where drops are impacted by micro-scale temperature/vapor fields).

20. lines 524-7: It seems to this reviewer that coalescence growth could also explain the "isolated large drop trend." Especially since the "this trend is associated with portions of the cloud where precipitation/condensate reaches the lowest altitudes from the respective cloud." This may be worth mentioning if you agree.

We are not sure this would be expected for collision-coalescence growth. Would this not be expected for the opposition trend? Namely, large drops would have a low likelihood of being significantly isolated from neighboring drops if a greater degree of collisions (and thus coalescence) is occurring? Or perhaps the reviewer then is referring to higher coalescence rates? Regardless, since there is no clearly expected trend related to collision-coalescence, we have opted to avoid mentioning this.

**Reviewer #2**

We appreciate the in-depth review provided here, and address the reviewer's comments below. We note that a prevailing theme of the review was a desire for increased clarity of the methodology, as well as other sections of the paper. Significant efforts were made to improve

the clarity of the manuscript, including editing the text as well as some tables and figures. We hope/believe these edits have improved the overall quality of the paper.

**Summary and Overall Evaluation**

This manuscript extends the existing literature that investigates sub-cm scale droplet clustering using data acquired by holographic imaging devices. The authors' primary conclusion is that within a few cloud trensects during the CSET campaign, HOLODEC data reveals a statistical tendency for large (> 25 µm) cloud drops to be spatially isolated. In addition, this tendency seems to be related to other bulk variables (e.g. low local number concentrations and sub-saturated conditions) which the authors argue is more consistent with entrainment-mixing than other bottleneck growth hypotheses.

In general, I am pleased to see the HOLODEC data from the CSET campaign further investigated; this is a very rich important dataset that – at least in this reviewer's opinion – has not been sufficiently discussed in the literature to date. The advantages of using instruments that detect 3-dimensional particle positions are clear and papers that extend existing techniques to more effectively leverage this information are important and needed. However, in this reviewer's opinion, there are significant improvements that are needed to make this work publish-able in ACP.

**Comment on Previous Review**

Although I read this paper several times prior to writing this review, immediately preceding writing this document I also glanced at the public review supplied by Anonymous Referee #1 (posted on October 8th). Since the task of responding to critical reviewer comments often involves trying to accommodate differing (and sometimes orthogonal) opinions, let me state from the outset that I am in agreement with all elements of the general assessment and major comments (and most of the minor comments) provided by this first Anonymous Referee. I believe addressing the concerns raised by that referee are necessary for publication of this manuscript and my additional comments below are meant to augment and extend those criticisms.

**Additional Major Comments**

1. The authors make the claim that the approach introduced here is novel and, though this particular implementation is new, the approach builds on previously published approaches that are not yet cited. (For example, the idea of building out statistics based on separation distance from an existing droplet has striking similarities to the conceptual starting point of the drop-size-dependent droplet clustering work using fractal and multi-fractal paradigms (see, e.g., [3, 8]). Further, the idea of using the local environmental conditions to characterize conditions under which the bottleneck problem may be better addressed was looked at in a way

with strong connections to this approach in [13]. Finally, the idea of looking at averaged-local environments and combining a variety of scales rather than a scale-localized measure like g (r ) has been explored repeatedly including through the use of the so-called volume-averaged pair-correlation function, Fishing statistic, Clustering index, Scaled Clustering Index, and through other common second-order statistics through the correlation-fluctuation theorem (see, e.g., [1, 2, 4, 5, 7, 11, 12]).

There are already several confusions associated with these tools and how they appropriately combine scaling information prevalent, but prevailing analysis suggests that aggregate statistics corresponding to a bulk statistic measured on scale R results from an unequally-weighted contribution from all scales less than or equal to R; the nature of this scaling, however, depends on the number of dimensions related to the observation, with different weighting factors existing in 1d, 2d, and 3d systems (see, e.g., [6]). Because of this, I am glad to see that the authors do not attempt to claim their approach identifies the key spatial scales of droplet clustering, but rather just detects whether individual drops are statistically isolated.

It is true that the majority of the above materials refer to analysis of optical-array-probe or similar 1-dimensional data, and thus the previous approaches have to be adapted to the three-dimensional domain – which is one of the contributions fo this manuscript. That being said, the approach outlined here very much is related to these previous works. To avoid confusion and to provide a bridge to previous work, I would encourage the authors to go back to those manuscripts to see if the language, notation, and approach can be adapted to align with already-existing frameworks.

Indeed, the authors' stated goal of "....determin[ing] the likelihood that drops of a given size will have a significantly high number of drops surrounding them as well as likelihoods that they are significantly isolated from neighboring drops" (lines 51-53) is very similar to the spirit of what the clustering index, Fishing statistic, volume-averaged pair-correlation function, etc. intend to do – determine if there is net clustering associated at or below some spatial scale. As the other reviewer suggested, the degree to which this approach is fundamentally different than a size-constrained version of previously-existing measures derived from and/or related to the radial distribution function is not clear. Although the authors of this manuscript suggest that the radial distribution function characterizes volumes, it calculates derived statistical quantities based on counting particle pairs separated by specified distances and thus every component of the sum in equation (1) ultimately is taken over an individual particle location. The stated different goal of doing this centered on a "per particle" basis vs. "per volume" basis claimed/implied by the authors may be a distinction without a true difference – especially as the metric utilized is ultimately found by adapting a single term of the sum utilized in calculating the RDF.

We agree with the reviewer's assessment here, and acknowledge that the chosen verbiage for the methodological novelty was in part because of the usage of the 3D holography rather than 1D measurements used in past literature as the reviewer mentions. We have gone through the

manuscript and made substantial changes through the text clarifying this point, while acknowledging similar methodologies used in the past.

To help elucidate our point that RDFs may present inherent biases, we now include additional text discussing these uncertainties in response to Reviewer #1 (referring to the response discussing Figure R1) as well as in part I of the Supplementary Material. This should highlight in part how our methodology differs from similar drop-by-drop methodologies (e.g., fishing statistic).

2. This is, in spirit, a continuation of the above point but it also bears mentioning that the work presented here also has links to existing work on nearest-neighbor and/or Voronoi analysis methods – especially in cases where the closer radial annuli have greater importance in the "local concentration/clustering" metric (as seems to be the case here if I am reading sections 2 and 3 correctly). This also links back to a different way of conceiving the radial distribution function as the sum of all of the kth nearest neighbor distributions (see, e.g, [9, 10]).

We acknowledge that our methodology of evaluating the most isolated drops for determine low DCFs is similar to a nearest-neighbor as well as Voronoi analysis. We have added discussion in the manuscript addressing this.

3. Despite rereading this paper multiple times (and being quite familiar with the general subject of droplet clustering statistics), all elements of figures 4, 5, and 6 are confusing to me. It seems like these figures provide a significant and necessary part of the central argument – but I cannot determine whether I am interpreting these results correctly. Figures 7 and 8 are slightly easier for me to comprehend, though I am still a bit stymied by how to interpret the uncertainty envelopes. The authors do attempt to describe all of these figures in detail, but I am unable to follow the explanations despite multiple attempts. I strongly suggest reconsidering whether there is an alternative way to visualize and/or explain these results that is more intuitive to a fresh reader.

We thank the reviewer for the assessment, and have made changes to these figures, as well as made significant changes to the text to improve its clarity. Concerning Figures 3-5, we add lines connecting the percentile datapoints in panels A,B,C,D. We also enlarge the bin center point in the middle zoomed in panel in Figure 3 to improve the clarity of how percentiles are determined. We hope these edits in combination with the extensive changes made in the text will be sufficient to improve the clarity of the analysis.

We have also made changes in the text to clarify how uncertainty is determined for Figures 6&7 in Appendix B:

Line 683-690: Uncertainty is determined using a permutation method where the same number of holograms are randomly selected from the combined HILD and OTHER simulations. For example, assume there are 300 holograms in the HILD category and 800 holograms in the

OTHER category. 300 holograms will be randomly selected from the combined HILD and OTHER simulations (which would equal 1200 if there are no shared holograms between the HILD and OTHER simulations) and classified as HILD. Likewise, 800 holograms will be randomly sampled from the combined simulations and classified as OTHER. This is done 10,000 times for each batch size, and the difference of these two distributions is determined for all 10,000 permutations. This produces a range of differences used to discern statistical significance.

4. To echo and extend a comment made by the first anonymous reviewer, the information and extensive set of logic and parameters presented in table 2 and table 3 are dizzying to a reader not yet fully comfortable with this analysis technique. The authors specifically talk about the desire to avoid analysis techniques that invite "p-hacking". The introduction of so many parameters associated with the technique raises the question as to whether or not the conclusions or quantitative results may be sensitive to the choice of these (what seem to be) arbitrary parameters associated with the method. For example – why 7 different radii? Why these cutoffs for N for ψ and shell size (table 2), why these cutoffs for HILD determination (table 3), why the batch sizes used (and would this matter if the instrument cadence were different like new developing holographic systems)? What here might depend on the number density of observations? What would change for clouds that are not stratocumulus or for larger fields-of-view or for two-dimensional PIV images? The claim is explicitly not that the authors chose parameters explicitly to get a significant result, but rather the (arguably equally important) point that to apply this technique more broadly to other campaigns, instruments, systems, or environments the authors provide little guidance to the reader how to adapt their methodology more broadly.

The choice of thresholds used for N in shell size and ψ were chosen to approximately produce terciles in the holograms such that ~1/3 of drops have low DCFs, ~1/3 of drops have high DCFs, and ~1/3 of drops meet neither category (in practice would possess the middle tercile of DCFs). Although mentioning it previously in the manuscript, we have rewritten it such as to emphasize its importance when deriving this ad hoc methodology.

Line 214-224: The remainder of this section outlines how individual droplets are diagnosed as being associated with significant clustering (those having high DCFs) or as being significantly isolated from neighboring drops (those having low DCFs), all of which is outlined in Table 2. This diagnosis will consist of two parts. The first part is to diagnose whether drops have a notable degree of high or low clustering. Specifically, in each hologram ~1/3 of the drops will be classified as having high DCFs and ~1/3 of the drops will be classified as having low DCFs (similar to Fig. 1). The second part will be to diagnose the degree of clustering for high DCFs and the degree of isolation for low DCFs. The first step is performed since the degree of clustering and isolation of droplets are computed in different ways, requiring droplets to first be separate into broad DCF categories (high DCFs and low DCFs). This first step is primarily discussed in Appendix A, and is shown as Step 3 in Table 2. This section focuses on diagnosing the degree of clustering and isolation of individual drops (Steps 4&5 in Table 2). Steps 1&2 in Table 2 show that $C_d$ is computed for all droplets, and highlights basic data filtering conditions.

*Perhaps most importantly, major results will be insensitive to minor variations in these ad hoc derived thresholds since the most notable results appear when analyzing the most isolated drops within the holograms (also noting there is no significant trend when analyzing drops within the most clustered regions [i.e., drops having the highest DCFs]). Therefore, thresholds can vary which will diagnose more or fewer significant DCFs within the holograms, but it will not make a difference when focusing on the drops having the highest and lowest DCFs within their respective holograms. This is demonstrated with a sensitivity test mentioned in Table S1 in the Supplementary material.*

*Broadly speaking, nearly all possible batch sizes are used, since the range of batch sizes results in the number of HILD and OTHER simulations approaching 0. This is seen in Figure B1A (in Appendix B), which shows the number of HILD simulations approach 0 for the smallest batch sizes (solid lines), and the number of OTHER simulations approach 0 for the largest batch sizes (dashed lines). In other words, the range of batch sizes captures nearly all possible cases where simulations can be classified as HILD or OTHER. However, the threshold of a batch size containing 100 simulations from both simulations does mean some batch sizes are removed.*

*We address the choice of 7 shell sizes when addressing reviewer 1: Namely, the choice of 7 is chosen in combination with conditions for classifying high and low DCFs (step 3 in Table 2) to approximately produce terciles of drops having high and low DCFs, while also providing adequate spatial resolution for discretizing drops to avoid producing similar degrees of high and low DCFs (for the purpose of sorting drops). We have simply updated the text to address this as:*

Lines 201-202: The number of shell sizes in our analysis is chosen in order to produce a reasonable resolution for determining spatial correlations between droplets.

To be explicit, my concern here is twofold: (i) the methods, though explained in detail, are confusing to the reader. Hopefully the general approach could be explained in a simpler manner and the finer details could be moved to an appendix or supplement? and (ii) as a reviewer who is tasked with validating the soundness
of the approach as well as the veracity of the results, there are just too many unjustified and (as far as I can tell) unphysically motivated parameters associated with the method to be sure that the conclusions are not a spurious consequence of the specific parameters used/chosen.

*We hope to have addressed this reviewer's concern in addressing the previous comment above. We have also made significant edits to the manuscript to improve upon the clarity of the methodology and simplifying its presentation as well. Table 2 is more straight-forward now, with no 'secondary' and 'additional' conditions which were previously present. We have also moved part of the methodology to a new appendix (Appendix A).*

5. One issue that may be of foundational concern – are we certain that the HOLODEC retrieval accuracy for droplets near an existing large droplet is equally effective as it would be in that region of the detector if the large drop were not present? HOLODEC reconstructions are built

upon an algorithm that very easily could have lower effectiveness for other droplet detections in the vicinity of a large drop, which could mean that the entirety of this analysis could be the result of a spurious instrumental anomaly. This concern isn't merely academic – although I am not an expert on the details of in-line holographic reconstruction, it seems entirely reasonable that – like in other optical reconstructions – a small signal near a large signal could be missed or falsely attributed to a fluctuation related to the large signal. Before making claims about the fundamental microphysics and precipitation initiation mechanisms, it seems quite important to ensure that the instrument has a quantum efficiency of small drop detection that does not depend on the presence or absence of a large drop nearby.

Given that it seems the statistical evidence is subtle, it bears pointing out that even if the sensitivity of small particle detection being impaired by the presence of a large particle nearby only affected part of the field of view that was retained for use in this study, that could very easily induce the entire statistical signal studied here.

We thank the reviewer for the comment. There is discussion in previous literature that such a bias can occur during the post-processing of holography, although admittedly little literature exists specifically for the HOLODEC. There is some literature that discusses postprocessing methods to avoid such biases (e.g., Fugal et al., 2009). Further, it only discusses its relevance for drop pair distances up to a couple hundred microns, given sufficiently large particles (often discussed for particles much larger than the maximum particle sizes of $D$=50 µm used in our analysis). The upper boundary of our first shell size (850 µm) is notably larger than this. We therefore suspect this is not an issue since holograms experiencing the "isolated large drop trend" possess relatively small drop concentrations (as discussed in the study; e.g., Figure 4), and these holograms with isolated large drops are commonly characterized as not having droplet pairs at distances up to 1500 µm. We suspect this bias would be best captured in high drop concentration holograms, where a greater number of droplets results in greater likelihoods of closer drop pair distances, and thus greater likelihoods of large drops "shadowing" or disturbing neighboring small droplet detection.

We also suspect that if this bias were prevalent, we would expect to observe it in all holograms rather than those only with low drop concentrations, those within subsaturated conditions (Figure 3), and those within select flight legs (Figure 5).

In addition, previous studies have found the HOLODEC to be in good agreement with commonly deployed single-scattering droplets probes such as the Fast Forward Spectrometer Probe (FFSP; Fugal and Shaw, 2009) and even the Cloud Droplet Probe evaluated during the CSET field campaign (Glienke et al., 2017).

Unfortunately, I don't believe the Monte Carlo mechanisms in place throughout this paper are designed in such a way as to verify or reject this concern. At the VERY least, in line 358, the authors should modify the text to read "...drops within the bottleneck size range are most likely to be measured to be significantly isolated from surrounding drops". However, I do not think that merely adding this language is sufficient to ignore this overarching concern. A skeptical

reader may be unconvinced by the entire message of the paper unless evidence that small particle detections near large particles are not inhibited in the holographic reconstruction.

In fact, we *do* believe the Monte Carlo-DCF analysis provides supporting information that this concern is less likely valid. First, we mentioned above how the isolated large drop trend is primarily observed in low drop concentration environments, which are less likely to have short drop pair distances and thus likelihoods of small drop detection interference in the presence of neighboring large drops. We also discussed how it's only observed for other select holograms (those in subsaturated conditions and those of select flight legs).

Second, this study shows that large drops are most likely to be isolated from nearby drops; and this is determined by looking at the most isolated drops and seeing bottleneck drops have a significantly high number of these low DCFs. The analysis also similarly evaluates the drops associated with the most clustering, and these results show drops with these significantly high DCFs are *no more or less likely to be large or small drops*. This is shown in Figure 3 for the red datapoints in the percentile panels (3A,B,D). Moving from panels 3A to 3B to 3D, percentiles show the likelihoods that drops having different sizes are more or less likely to be associated with increasingly high DCFs. Even for droplets associated with the most clustering (highest DCFs), large drops are not more or less likely than small drops to possess such DCFs.

In other words, the lowest DCFs are more likely to be possessed by bottleneck drops rather than smaller drops, whereas the highest DCFs are not associated with any increases/decreases in being possessed by any given drop size. If the droplet interference were to be prevalent, then we would also expect to see this trend associated with drops having high DCFs. Specifically, we would expect to see bottleneck drops are significantly unlikely to possess high DCFs (e.g., we would expect to see red diamonds and/or stars for large drops near percentiles approaching 0 in Figures 3A,B,D).

While we understand the reviewer's concern, we believe that the language in line 358 already implies that drops within this size range "are measured to be" significantly isolated from surrounding drops, since we are dealing with measurements. However, we have included new discussion in Appendix C: Uncertainties and Sensitivity Tests addressing the reviewer's concern.

Line 703-708: Errors may also be associated with droplets producing diffraction patterns that disturb those from closely neighboring drops, which may limit the detection of small drops within close proximity of larger drops. However, we suspect this is not a concern since the isolated large drop trend is primarily observed within holograms having low drop concentrations, where droplet pair distances are generally greater than for holograms having high drop concentrations.

**Additional Minor Comments**

1. Although this is a minor comment, I believe it is an important one. Much of the work over the last 20+ years on cloud particle clustering has been obfuscated because of the lack of a consensus and author-to-author consistency regarding terminology. Tying what has been done to previous work is particularly important (which is the foundation underlying my first two major comments), but taking care to do so in a way that doesn't further confuse things is also important. As such, I implore the authors to choose a different notational convention than to use the idea of "$g_{drop}$" (e.g. see equation (3)). Multiple papers already have been written about the conditions as to when g(r) can and cannot be calculated and still contain the physical meaning ascribed to it, and many of those conditions are not easily mappable to a particle-by-particle approach as presented here. (For example, although g(r) can always be calculated from any existing data-set, the underlying assumption of spatial homogeneity requires $g(r \to \infty) \to 1$ and statistical viability requires the denominator to have sufficient expectation so that shot-noise doesn't dominate the signal. Some of these concerns may not persist for individual particle detections, but assuredly if the whole point is to determine whether a particle is statistically isolated the premise of the approach is that there is spatial heterogeneity on the scales of investigation. Although it may seem harmless to adopt the use of the statistical metric outside its formal range of validity, we have previously seen (see, e.g, [1] and the corresponding counter-examples presented in [7] that applying radial distribution functions outside of the range of validity for homogeneous systems results in misunderstandings.) The use of g (r) on a particle-by-particle metric invites further confusion, especially if the approach introduced here gains wider traction.

We understand the reviewer's concern and have updated $g_{drop}$ to be $C_d$ so as not to cause confusion as discussed by the reviewer.

2. I would find it valuable if the total number of holograms were presented for each of the flight legs in table 1 as well, so the reader can get a sense of the accepted/utilized fraction of detected holograms in each of the flight legs.

We have now included the total number of holograms, with the only condition being that the hologram's outer sample volume (0.3x0.3x10 cm) exceed 100 (as discussed in the text).

3. The paper cited on line 109 by Thiede et al. does not use HOLODEC data. (The system used in that paper has some similarities to HOLODEC, but is mounted on an aerostat, takes observational data at a much higher cadence, and is processed slightly differently).

We appreciate the reviewer pointing this out and have updated the text accordingly

Line 114-117: Whereas previous studies have quantified droplet clustering of the HOLODEC's respective sample volumes (Glienke et al., 2020; La et al., 2022; Larsen et al., 2018; Larsen and Shaw, 2018;) or the Advanced Max Planck CloudKite holography system's measurement volumes which are similar to the HOLODEC (Thiede et al., 2025), results here evaluate drop clustering on a drop-by-drop basis.

4. Given the fairly severe aspect ratio utilized (see line 119), there are questions as to the true three-dimensionality of the domain explored here. It is notable that this has a more extreme aspect ratio than other studies that have used the HOLODEC data, which is one of the reasons that the guard-area (or, as stated here, guardrail) approach has been largely abandoned in some other work. It is not clear how that may impact the final results.

We note that one of the sensitivity tests performed was to extend the guardrail distance to include more droplets in the analysis (Supplementary Table S1). We include the results below which extends the dimensions of the guardrail box to be 0.4x0.4x9.8 cm (compared with 0.3x0.3x9.7 cm in the manuscript) and the outer volume to 0.7x0.7x10.1 cm.

[Figure]

Figure R6: Similar to Figure 3 in the manuscript, but using a larger guardrail volume (0.4x0.4x9.8 cm) compared to the one used in the manuscript.

Note that all major trends are similarly observed when extending the guardrail volume. Namely, the most isolated drops (those having low DCFs) are significantly likely to be large drops. This can be seen when comparing modestly isolated drops (panel C) with the most isolated drops (panels B and D).

5. Line 143 – I don't believe the authors meant to say i th particles here.

This was intentional. The $i$th particle corresponds to $i$ defined in eq 1.

6. Line 165 – "counting statistic error" – I'm not sure that "error" is the appropriate term in this context. "Uncertainty" or "bias" or "fluctuation" may be more appropriate here.

Changed to "bias"

7. Line 179 – "maximum" probably should be "minimum" here.

Updated

8. Lines 208-210 – see major comment 2 related to similarities/links to Voronoi/nearest-neighbor techniques.

We agree the method is comparable to Voronoi/nearest neighbor techniques. We now mention this:

Line 240-242: Selecting the maximum shell sizes containing no neighboring drops is comparable to using a Voronoi or nearest neighbor approach for diagnosing isolated droplets. However, a simpler approach is used here since we simply wish to diagnose degrees of drop isolation relative to the holograms they are contained in.

9. Line 222 – In what way is 0.15 cm "ideal"? Would that parameter extend to other cloud conditions/instruments/etc? This links back to major comment 4.

The original sentence intended to highlight the trade-offs for selecting larger shell sizes. Namely, while larger shell sizes allow for determining increasingly isolated drops, they also require shrinking the guardrail sample volume. Thus, it is solely related to the instrumentation. We edited this section to avoid using the term "ideal".

Line 251-254: Utilizing larger shell sizes allows for the ability to diagnose increasingly isolated drops. However, larger shell sizes consequently decrease the guardrail volume, which limits the available sample volume (depicted as $r + \delta r$ in Fig. 2). The upper bound of 0.15 cm is chosen judiciously to diagnose a notable degree of separate between drop pairs while also obtaining a sufficient sample of drops within the bottleneck size range.

10. Line 243 – By choosing the same number of high and low DCFs to control for environmental conditions also seems like it introduces another scale parameter for the analysis, especially if this method were to be applied to a more heterogeneous domain in regards to number concentration or other parameters.

We argue the results show minimal dependence on this methodological component. This is because we evaluate trends by selecting a different number of high and low DCFs, and notable trends are observed when examining the highest and lowest DCFs. For example, if we simply had a method to classify 50% of DCFs as high and 50% of DCFs as low, notable findings in the study are still primarily restricted to the highest and lowest DCFs. Thus, our study primarily focuses on the most isolated drops, rather than all/most drops that are modestly isolated.

However, the reviewer is correct that this may impact the weighting of different environmental conditions. For example, holograms with higher drop concentrations will consequently possess more drops classified as having high and low DCFs compared with holograms having low drop concentrations. However, our Monte Carlo-hologram comparison methodology avoids this problem by randomly selecting holograms to analyze (Section 4.2), which the major findings of this study ultimately come from.

11. Line 246 – "randomly" – I'm not sure what is meant by this in this context.

We've updated the text for clarity:

Line 266-267: To achieve this, the DCF category with the greater number of drops is reduced to have the same number of drops as the lower category. Drops are removed at random to produce an equal number of drops between the DCF categories.

12. Lines 470-473. The fact that holograms with Dmax > 45 μm do not exhibit this trend is very much concerning and, to me, seems to run counter to the authors' explanation that this signal is likely induced by entrainment-mixing.

Two points:
1. There are only 96 holograms containing $D_{max}$ 45–50 μm (our max drop size analyzed is 50 μm), meaning the monte carlo method may not have a sufficient sample size in order to capture an isolated trend for these relatively larger drops. This is noteworthy considering the methodology of randomly sampling, particularly for batch sizes far exceeding 96 holograms.
2. We mentioned in the additional comments section (Section 5) that a diffusional growth mechanism may only occur up to a given size. We do not claim that drop sizes may grow beyond this size due to the speculated entrainment-mixing growth mechanism.

13. Lines 525-527. The fact that holograms with larger drops exhibit broader size distributions and fluctuations of other microphysical variables is more or less tautologically true, isn't it? (Especially if the instrument can only see droplets above a certain size reliably.)

Perhaps strongly related but we would argue not tautologically true. The max drop size only considers one drop in a hologram, rather than all drops when determining the spread of drop sizes. Also, the increase in mean diameter could correspond with broader drop size

distributions (partially evaporating the relatively small drop population) or narrower drop size distributions (completely evaporating the relatively small drop population).

14. Line 550 – at the end of the sentence including numbers, it seems to be important to specify that these are relative to the drop's local environment and not in absolute/cloud-averaged terms. This local environment is tied to a specific measurement scale, and that is another spatial scale of relevance when exploring this question.

We thank the reviewer for pointing this out and have updated the text accordingly

15. Line 554 – "significantly isolated"; this raises the question as to whether being statistically significant is sufficient to affect localized dynamics/microphysical responses.

To avoid the ambiguity of "significantly", since its use is context dependent on the statistical methodology used here, we have rephrased this sentence:

Line 561-562: Results here show drops having sizes within the bottleneck size range (D~25–50 µm) are commonly the most isolated drops within their respective holograms.

16. Line 570 – I believe it is important at this stage to indicate the scale (sub-cm scale) where this conclusion is likely valid.

We agree with the reviewer. However, we note that the very next sentence states that this analysis focuses on sub-cm scale clustering (provided below). We opted to leave it as is, and hope this is acceptable.

An additional notable departure from these past studies is that the particle-by-particle clustering is only diagnosed on spatial scales of millimeters, whereas those studies also consider centimeter scale drop spatial inhomogeneities when diagnosing clustering.

Bibliography

Fugal, J. P. and Shaw, R. A.: Cloud particle size distributions measured with an airborne digital in-line holographic instrument, Atmospheric Measurement Techniques, 2, 259–271, https://doi.org/10.5194/amt-2-259-2009, 2009.

Fugal, J. P., Schulz, T. J., and Shaw, R. A.: Practical methods for automated reconstruction and characterization of particles in digital in-line holograms, Meas. Sci. Technol., 20, 075501, https://doi.org/10.1088/0957-0233/20/7/075501, 2009.

Glienke, S., Kostinski, A., Fugal, J., Shaw, R. A., Borrmann, S., and Stith, J.: Cloud droplets to drizzle: Contribution of transition drops to microphysical and optical properties of marine stratocumulus clouds, Geophysical Research Letters, 44, 8002–8010, https://doi.org/10.1002/2017GL074430, 2017.

Larsen, M. L. and Shaw, R. A.: A method for computing the three-dimensional radial distribution function of cloud particles from holographic images, Atmospheric Measurement Techniques, 11, 4261–4272, https://doi.org/10.5194/amt-11-4261-2018, 2018.

Larsen, M. L., Shaw, R. A., Kostinski, A. B., and Glienke, S.: Fine-Scale Droplet Clustering in Atmospheric Clouds: 3D Radial Distribution Function from Airborne Digital Holography, Phys. Rev. Lett., 121, 204501, https://doi.org/10.1103/PhysRevLett.121.204501, 2018.